# Developmental trajectories of EEG aperiodic and periodic components in children 2–44 months of age

Carol L. Wilkinson [1,2] ✉, Lisa D. Yankowitz[1,2], Jerry Y. Chao [3], Rodrigo Gutiérrez[4,5], Jeff L. Rhoades [6,7], Shlomo Shinnar[8,9], Patrick L. Purdon[5] & Charles A. Nelson[1,2,10]

The development of neural circuits has long-lasting effects on brain function, yet our understanding of early circuit development in humans remains limited. Here, periodic EEG power features and aperiodic components were examined from longitudinal EEGs collected from 592 healthy 2–44 month-old infants, revealing age-dependent nonlinear changes suggestive of distinct milestones in early brain maturation. Developmental changes in periodic peaks include (1) the presence and then absence of a 9-10 Hz alpha peak between 2-6 months, (2) nonlinear changes in high beta peaks (20-30 Hz) between 4-18 months, and (3) the emergence of a low beta peak (12-20 Hz) in some infants after six months of age. We hypothesized that the emergence of the low beta peak may reflect maturation of thalamocortical network development. Infant anesthesia studies observe that GABA-modulating anesthetics do not induce thalamo-cortical mediated frontal alpha coherence until 10-12 months of age. Using a small cohort of infants ($n = 23$) with EEG before and during GABA-modulating anesthesia, we provide preliminary evidence that infants with a low beta peak have higher anesthesia-induced alpha coherence compared to those without a low beta peak.

The infant brain undergoes dramatic structural and physiological change in the first year after birth. Rapid increases in brain volume coincide with expansive synaptogenesis[1–3], as well as interneuron migration, maturation, and network integration[4]. In particular, during this early period thalamocortical connections are established through an intricate sequence that plays a critical role in the development of sensory cortical networks[5]. However, the detailed timing of interneuron and thalamocortical maturation in human development is largely unknown. In rodent models, the development of thalamocortical circuitry is notable for transient inhibitory connections that drive subsequent circuit formation and coincide with critical periods of plasticity present during the first 2–3 postnatal weeks[6]. In humans, longitudinal resting-state fMRI data suggest that while thalamic-sensorimotor connectivity networks are present at birth, other networks (e.g., thalamus-medial-visual, thalamus-default-mode) do not emerge until 1 year of age[7]. However, MRI studies thus far have been limited to measuring annual changes in structural or functional connectivity, preventing a detailed understanding of rapid

[1]Division of Developmental Medicine, Boston Children's Hospital, Boston, MA, USA. [2]Harvard Medical School, Boston, MA, USA. [3]Department of Anesthesiology, Montefiore Medical Center, Children's Hospital at Montefiore, Albert Einstein College of Medicine, Bronx, NY, USA. [4]Departamento de Anestesia y Medicina Perioperatoria, Hospital Clínico de la Universidad de Chile, Santiago, Chile. [5]Department of Anesthesia, Critical Care and Pain Medicine, Massachusetts General Hospital, Boston, MA, USA. [6]Department of Neurobiology, Harvard Medical School, Boston, MA, USA. [7]Program in Neuroscience, Division of Medical Sciences, Graduate School of Arts and Sciences, Harvard University, Cambridge, MA, USA. [8]The Saul R. Korey Department of Neurology, Montefiore Medical Center, Albert Einstein College of Medicine, Bronx, NY, USA. [9]Department of Epidemiology and Population Health, Albert Einstein College of Medicine, Bronx, NY, USA. [10]Harvard Graduate School of Education, Cambridge, MA, USA. ✉e-mail: carol.wilkinson@childrens.harvard.edu

developmental change during this period. In contrast, electroencephalography (EEG) can provide frequent and non-invasive repeated measurement of brain oscillations that directly result from transient developmental changes in inhibitory networks and maturation of thalamocortical circuitry[5,8,9].

The EEG power spectrum is comprised of two physiologically distinct components reflecting underlying neuronal activity: aperiodic and periodic activity. The aperiodic component defines the slope of the power spectrum, following a 1/f power law distribution (Fig. 1a) and reflects non-oscillatory neuronal spiking activity[10–12]. In addition,

recent evidence suggests that the aperiodic slope reflects the excitatory-inhibitory (E/I) balance of the underlying neuronal network, where a flattened, reduced slope is associated with increased excitation over inhibition, and a steeply more accelerated slope with increased inhibition over excitation[11]. Longitudinal studies extending from childhood to adulthood have observed decreases in aperiodic slope with age, suggestive of increases in E/I balance with age[13–16]. Changes in the aperiodic component in early infancy are less well described, and we hypothesize they may be substantially different from those in childhood, as the first year after birth includes rapid

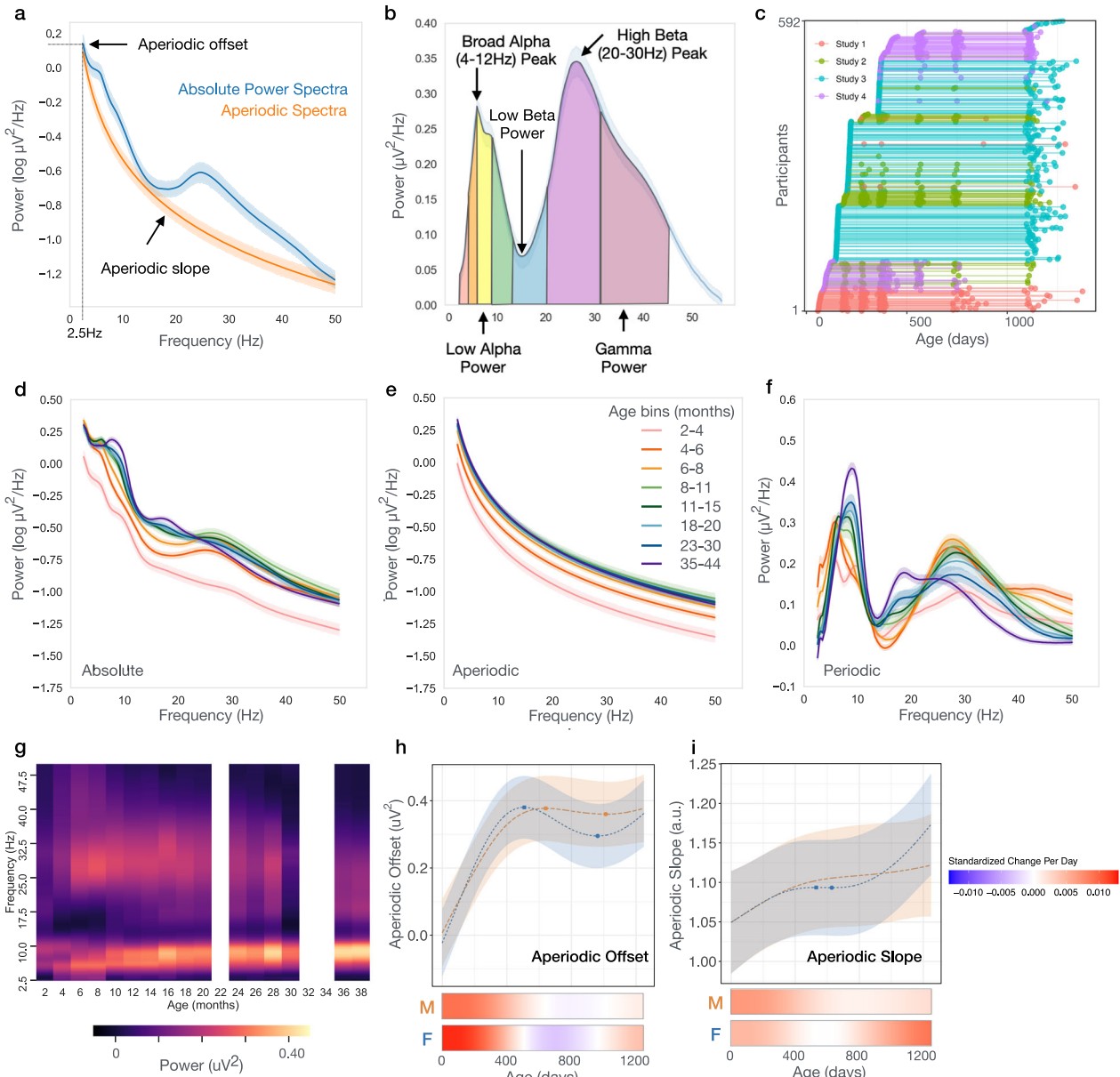

Fig. 1 | **Developmental trajectories of aperiodic and periodic power spectra.** **a**, **b** Example power spectra derived from 6–8 month-old participants from Study 2. Power spectra are represented as the mean value with shaded errors describing 95% confidence intervals. **a** Aperiodic offset is defined as power at 2.5 Hz. **b** Periodic power peaks defined as maxima within a defined frequency range. Periodic band power defined as the integral of the periodic power spectra between defined frequencies. **c** Longitudinal study enrollment. Each line is a participant with dots indicating when EEG was collected for that participant. **d**–**f** Absolute, Aperiodic, and Periodic power visualized across 8 age bins (see Table 1). Spectra from EEGs collected within each age bin were averaged and shading describes 95% confidence intervals. **g** Heatmap showing age-related changes in periodic power binned every 2 months. **h**, **i** GAMMs modeled trajectories of aperiodic offset and slope for males (orange) and females (blue). Lines are the model predicted value with the shaded area representing 95% confidence intervals. Here, age is incorporated into the model using exact age in days, rather than age-bins. Relative inflection points are shown with circular markers. Source data are provided as a Source Data file. Below, heatmaps show the standardized change in offset or slope per day, defined as [change per day]/[standard deviation of measures across full age range].

increases in neuronal activity, synaptogenesis, and inhibitory neuron integration.

The periodic component of the power spectrum is defined as the portion of the absolute power spectrum rising above the aperiodic slope (Fig. 1b). Periodic power reflects oscillatory activity occurring in narrow frequency bands that are highly correlated with various cognitive processes and behavioral states[17,18], and provide the foundation for both local and long-range communication within the brain. The majority of neural oscillations observed in the power spectrum are the direct result of inhibitory and thalamocortical network responses to sensory input[19–23]. Thus, as a measure, the EEG power spectrum is well positioned to shed light on the developmental timing of inhibitory and thalamocortical network maturation.

Thus far, developmental EEG studies have largely focused on theta/alpha oscillations which are modulated by thalamocortical interactions and are associated with cognitive functions of attention and memory[24–26]. Multiple studies of the first two years of life have observed a shift in peak frequency from a 5 Hz theta peak at 5 months to 8 Hz alpha peak at 2 years, coinciding with increases in alpha power across this period[27–31]. This dominant peak frequency continues to increase into the mature 10 Hz posterior alpha rhythm by adolescence[9,32]. It is hypothesized that the gradual shift in dominant peak frequency is modulated by maturation of thalamocortical circuitry in concert with developmental gains in cognitive functions[9]; however, the precise mechanisms remain unknown.

Unlike theta/alpha power, little is known about the early developmental changes in periodic beta power. In adults, beta oscillations are strongly associated with sensorimotor processing in addition to higher-order cognitive tasks such as working memory[33]. Similar to alpha oscillations, the generation of beta oscillations relies on GABAergic interneuron networks and thalamocortical connectivity[34–36]. In adults, low-dose GABA-modulating anesthetics induce a sedative state with 13–25 Hz beta oscillations, whereas higher doses used to maintain unconsciousness progressively slow these beta oscillations into coherent, frontal specific, alpha oscillations[36–38]. However, GABA-dependent anesthesia-induced frontal alpha coherence does not emerge in infants until after 10 months of age and is not consistently present until 15–20 months of age[39–41]. Anesthesia-induced alpha coherence is hypothesized to involve GABA-dependent thalamocortical loops leading to hypersynchronization between thalamic and prefrontal cortices[38,42,43]. Simultaneous recordings of thalamic nuclei and cortices performed in both rodent and monkeys during propofol administration have demonstrated anesthesia-induced alpha coherence between structures[42,43]. Therefore, potential covariation of developing beta oscillations and anesthesia-induced counterparts may lend insight into the role and time course of developing inhibition in human thalamocortical circuit development.

Using longitudinal EEG data collected from 592 healthy infants (yielding a total of 1335 EEGs) from 2 to 44 months after birth, in this work we characterize early developmental trajectories of EEG aperiodic components and periodic power from 2 to 50 Hz. Consistent with known increases in brain volume and synaptogenesis, we observe rapid increases in aperiodic offset in the first year, with minimal change between 1 and 3 years[1–3]. In addition, we observe transient periodic peaks in alpha power at 2–3 months and high beta power at 4–18 months. A low beta peak (12–20 Hz) also begins to emerge in infants starting as early as 6 months of age. We hypothesized that emergence of this low beta peak reflects maturation of early connections between the thalamus and cortex. To test this hypothesis, we leveraged a smaller dataset consisting of a cohort of infants with EEG recordings before and during clinical anesthesia. Consistent with our hypothesis, we find infants with an identifiable low beta peak have higher anesthesia-induced alpha coherence compared to those that who do not, providing preliminary evidence

that the emergence of this peak may be associated with thalamo-cortical loop maturation.

## Results

Resting-state EEG were collected longitudinally from 592 healthy infants, aged 2–44 months, across 4 studies occurring in the same laboratory (Fig. 1c and Table 1). Whole brain power spectra for each individual were calculated using a multitaper spectral analysis[44] for each electrode, and then averaged across electrodes (Supplemental Fig. 1. Individual spectra shown in Supplemental Fig. 2). For visualization of developmental changes, spectra were then averaged across individuals within 8 age bins (Fig. 1d). Notable nonlinear changes in aperiodic and periodic power spectra were observed between age bins, including transient peaks in the periodic spectrum across both alpha and beta frequency ranges (Fig. 1e–g). To further characterize these developmental changes in the spectra, we used generalized additive mixed models (GAMMs) to model nonlinear trajectories of power parameters. For each model an age-by-sex interaction was tested for significance. If not significant, the interaction term was removed, and the model was refit using sex as an additive covariate. All models also included random effects of study and participant. Spaghetti plots of individual non-GAMMs modeled data are shown in Supplemental Materials.

**Table 1 | Sample Characteristics**

| | Combined studies | Study 1 | Study 2 | Study 3 | Study 4 |
|---|---|---|---|---|---|
| | N = 592 | N = 49 | N = 72 | N = 363 | N = 108 |
| **Sex, % female (n)** | 46.3 (274) | 51.0 (25) | 45.8 (33) | 46.3 (168) | 44.4 (48) |
| **Ethnicity, % (n)** | | | | | |
| Hispanic | 8.6 (51) | 16.3 (8) | 1.4 (1) | 10.5 (38) | 3.7 (4) |
| Non-Hispanic | 90.2 (534) | 81.6 (40) | 97.2 (70) | 88.4 (321) | 95.4 (103) |
| Not answered | 1.2 (7) | 2.0 (1) | 1.4 (1) | 1.1 (4) | 0.9 (1) |
| **Race, % (n)** | | | | | |
| White | 74.0 (438) | 10.2 (5) | 86.1 (62) | 79.1 (287) | 77.8 (84) |
| Black or African-American | 6.4 (38) | 53.1 (26) | 1.4 (1) | 2.2 (8) | 2.8 (3) |
| Asian | 3.9 (23) | 4.1 (2) | 2.8 (2) | 4.1 (15) | 3.7 (4) |
| More than one race | 12.8 (76) | 14.3 (7) | 8.3 (6) | 12.9 (47) | 14.8 (16) |
| Other | 1.4 (8) | 12.2 (6) | 0 | 0.6 (2) | 0 |
| Not answered | 1.5 (9) | 6.1 (3) | 1.4 (1) | 1.1 (4) | 0.9 (1) |
| **Family income, % (n)ª** | | | | | |
| <$35,000 | 5.4 (32) | 32.7 (16) | 4.2 (3) | 2.5 (9) | 3.7 (4) |
| $35,000–$75,000 | 11.1 (66) | 18.4 (9) | 9.7 (7) | 13.2 (48) | 1.9 (2) |
| >$75,000 | 73.8 (437) | 18.4 (9) | 70.8 (51) | 76.3 (277) | 92.6 (100) |
| Not answered or do not know | 9.6 (57) | 30.6 (15) | 15.3 (11) | 8.0 (29) | 1.9 (2) |
| **Participant EEG data included in analysis, n (% with longitudinal data)** | | | | | |
| 2–4 m | 97 (91) | 46 (93) | 10 (100) | – | 41 (85) |
| 4–6 m | 119 (46) | – | 2 (100) | 113 (45) | 4 (50) |
| 6–8 m | 223 (68) | 43 (100) | 50 (100) | 130 (45) | – |
| 8–11 m | 95 (98) | 37 (100) | 58 (98) | – | |
| 11–15 m | 291 (79) | 40 (100) | 62 (100) | 102 (45) | 87 (95) |
| 18–20 m | 107 (97) | – | 40 (100) | – | 67 (95) |
| 23–30 m | 118 (99) | 22 (100) | 44 (100) | – | 52 (98) |
| 35–44 m | 285 (92) | 18 (100) | 48 (95) | 174 (90) | 45 (95) |
| By participant, mean # EEGs ± SD | 3.2 ± 1.7 | 4.6 ± 1.1 | 4.9 ± 1.3 | 1.6 ± 0.5 | 3.2 ± 0.9 |

ª2 infants reported to have income of $30–39,000 were placed in <35,000 category. 1 infant with reported income of $70–79,000 placed in >75,000 category.

## Aperiodic activity increases most during first year of life

First, we assessed age-dependent changes in the aperiodic component and observed the largest developmental increases in aperiodic activity between 2 and 8 months after birth (Fig. 1e). Aperiodic offset and slope significantly increased with age (FDR-adjusted $q$ values < 0.001), and age-by-sex interactions were present for both aperiodic offset ($F = 5.28$, $q = 0.002$) and slope ($F = 3.04$, $q = 0.03$). Modeled developmental trajectories of the aperiodic offset showed a qualitatively sharp linear increase over the first year after birth for both males and females (Fig. 1h). Modeled developmental trajectories of the aperiodic slope showed a qualitatively gradual increase over the first year. These findings contrast with consistent reports of decreasing offset and slope across child and adulthood[13–16], and likely reflect the known increases in brain volume and synaptogenesis occurring across the first year of life. Differences in developmental trajectories between 4 regions of interest (ROI) (frontal, central, temporal, and posterior) were also assessed (Supplemental Figs. 5 and 6). The posterior ROI had higher aperiodic offset than all other ROIs, with the greatest increase in offset occurring in the first year (Frontal $F = -32.47$, $q < 0.0001$, Central $F = -50.12$, $q < 0.0001$, Temporal $-39.29$, $q < 0.0001$; Supplemental Fig. 3B).

## Transient 9.5 Hz alpha peak observed in 2–4 month-old infants

At the youngest age bin (2–4 months) two peaks with similar amplitude are observed across the theta/alpha (4–12 Hz) range in the majority of infants (69%; Fig. 2a, c). A lower frequency peak is observed in the theta (4–6 Hz) range at $5.5 \pm 0.3$ Hz, and higher frequency peak is observed in the alpha (6–12 Hz) range at $9.5 \pm 0.45$ Hz. However, by 6-months only 15% of infants have two peaks in this range, and for most infants it is the higher 9.5 Hz peak that is no longer observed. At 6 months, fewer than 40% of infants exhibit a dominant peak in the "alpha" (6.5–12 Hz) range (Fig. 1d) and the average peak frequency in the theta/alpha range is $6.3 \pm 1$ Hz. Age (in days) was negatively associated with the probability of having two peaks across the 2–6 month age range in a generalized linear mixed effects model (odds ratio = 0.98, $B = -0.018$, $p < 0.0001$). This disappearance of the higher peak after 4-months of age may reflect a transient step in thalamocortical circuit development. Previous research has observed a gradual shift in peak frequency from 5 to 8 Hz from infancy to early childhood, however these studies started no earlier than 5 months of age[27–30]. In order to assess whether an increase in peak frequency beginning at 5 months is present in our data set we modeled developmental trajectories of peak amplitude and frequency between 4 to 12 Hz starting at 170 days, when the vast majority of EEGs exhibited a single dominant peak. No age-by-sex interactions were observed in GAMMs modeled trajectories, and consistent with previous studies peak frequency and peak amplitude significantly increased with age (frequency: $F = 138.3$, $q < 0.0001$ Fig. 2e; amplitude: $F = 416.62$, $q < 0.0001$, Fig. 2f). Figure 2g–i show modeled trajectories for EEG power calculated over defined frequency bands commonly used in infant EEG research: theta (4–6 Hz), low alpha (6–9 Hz), and high alpha (9–12 Hz). An age-by-sex interaction was present for theta power, although qualitatively the shapes of trajectories were similar (Fig. 2g; $F = 5.5$, $q < 0.01$).

## Transient beta peaks between 4 and 18 months

Several age-dependent transient changes are observed in the low beta (13–20 Hz) and high beta (20–35 Hz) range. First, the shape of the periodic power spectra in the low beta range is notable for the absence of a low beta peak prior to 1 year of age (Fig. 3a), with only 10% of infants (24/222) exhibiting a peak between 6 and 8 months of age (Fig. 3a–c). After 8 months, a low beta peak begins to emerge in some of infants, with 48% (52/107) showing a peak at 18–20 months, and 70% (199/285) by 36 months (Fig. 2c). As a low beta peak was not identified in many children across the age range, peak amplitude and frequency was not modeled. In contrast, virtually all (99.5%) of the infants had an

identifiable high beta peak prior to 12 months of age (Fig. 3a). However, notable nonlinear shifts in frequency and amplitude of the high beta peak were observed (Figs. 1f and 3d, e). During the first year after birth, the high beta peak amplitude increases, peaking at 372 days (12.2 months), and then substantially decreases until 1021 days (2.8 years). High beta peak frequency trajectories are also nonlinear, with peak frequency at its highest at 491 days (male 28.4 Hz, female 28.8 Hz), followed by a steady decline in frequency. Modeled trajectories of periodic power for commonly used frequency bands are shown in Fig. 3f–h: low beta (12–20 Hz), high beta (20–30 Hz), and gamma (30–45 Hz).

The observed nonlinear changes across the beta range are striking. While many EEG infant studies group beta oscillations into a singular frequency range, the data presented here supports that low and high beta have distinct developmental origins. Specifically, between 6 and 24 months we observe the gradual emergence of a low beta peak, and simultaneously the rise and fall of a prominent high beta peak, ultimately resolving into a broader beta peak by 36 months.

Traditionally, beta oscillations measured in children and adults are associated with sensory and motor processing, where reductions in beta power are observed during the preparation or execution of motor tasks[33]. However, beta activity has also been shown to be modulated during a wide range of nonmotor cognitive tasks[33,45]. The developmental emergence of low beta oscillations may represent sensorimotor skills (e.g., crawling, walking) gained during this period, but may also represent the developmental maturation of neural circuitry. For example, GABAeric interneuron networks and thalamocortical connectivity are highly associated with the generation of cortical beta oscillations, as well as anesthesia-induced frontal alpha coherence, but neither are not fully established at birth[39].

## Low beta peak associated with anesthesia-induced alpha coherence

We wondered whether developmental changes in infant beta power measured in a resting state may represent concurrent maturation of GABAergic interneuron networks and thalamocortical connectivity. Multiple lines of evidence suggest that anesthesia-induced frontal alpha coherence is dependent on thalamocortical connectivity[38,42,43], and in infants robust levels of alpha coherence with anesthesia administration are not observed until 10 months of age[40]. We hypothesized that the emergence of low beta oscillations (as measured by the presence of a low beta peak) beginning after 7-months of age may reflect maturation of the thalamocortical loop also responsible for the developmental emergence of anesthesia-induced frontal alpha coherence around the same age[40]. To explore this possibility, we analyzed data from infants participating in cross-sectional study where EEG data was collected before and during exposure to GABA-modulating sevoflurane anesthesia[46]. All infants were undergoing elective procedures (e.g., circumcision) and infants were excluded for prematurity, neurologic injury, epilepsy, or planned intracranial surgery. We then tested whether infants with a low beta peak during the awake, unanesthetized state had increased GABA-dependent anesthesia-induced frontal alpha coherence. EEG data from 36 infants across a broad age range (6–15 months old) were collected during the (1) awake pre-anesthetized and (2) anesthetized states. We first analyzed the baseline awake state (pre-anesthesia) EEG across all 36 infants to confirm that similar developmental changes were observed. Developmental changes in the periodic power spectra in this smaller dataset were qualitatively similar to those described above (Fig. 4a), with a low beta peak beginning to emerge in some infants after 7 months and present in roughly half the infants between 7 and 12 months of age (11/23). To test our above hypothesis, we then limited our analysis to this 7–12 month age range ($n = 23$), and compared alpha coherence during anesthesia in those with and without a low beta peak in the awake, unanesthetized state.

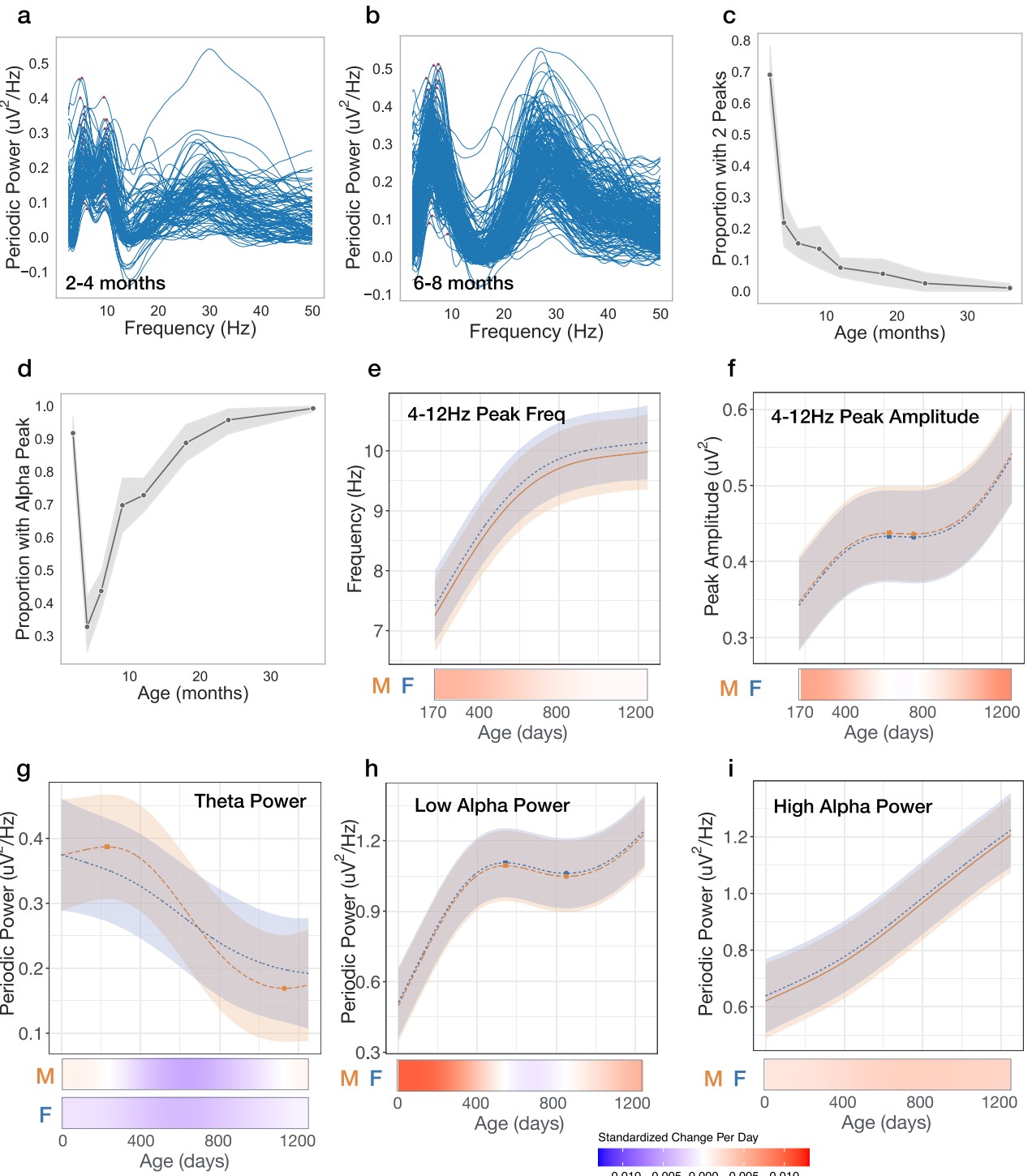

**Fig. 2 | Transient and nonlinear changes in periodic power between 4 and 12 Hz.** **a**, **b** Individual periodic power spectra for 2–4 months, and 6–8 months old. Red markers show peaks between 4 and 12 Hz. **c** Proportion (mean of binary data) of infants with two peaks identified between 4 and 12 Hz at each age bin. **d** Proportion of infants with an identified peak between 6.5 and 12 Hz at each age bin. For c and d, shaded areas represent 95% confidence intervals. **e**–**i** GAMMs modeled trajectories for males (orange) and females (blue). Lines are the model predicted value with the shaded area representing 95% confidence intervals. Relative inflection points are shown with circular markers. Source data are provided as a Source Data file. Below, heatmaps show the standardized change in offset or slope per day, defined as [change per day]/[standard deviation of measures across full age range]. Both male and female heatmaps shown for models with significant age × sex interaction.

As hypothesized, both ANCOVA and a generalized linear mixed effects (LME) model demonstrated a significant effect of low beta peak presence on median alpha coherence, with increased alpha coherence in those with a low beta peak compared to those without (ANOVA $F(1,20) = 5.25$, $p = 0.03$; LME $\beta = 0.12$, SE = 0.051, $p = 0.02$; Fig. 4B).

## Discussion

We present longitudinal analysis of a large sample EEG data collected between 2 and 44 months of age. Findings provide insight into the developmental timing of inhibitory network and thalamocortical circuit maturation during human infancy. Several age-dependent findings in our study contrast with previous longitudinal studies of child and

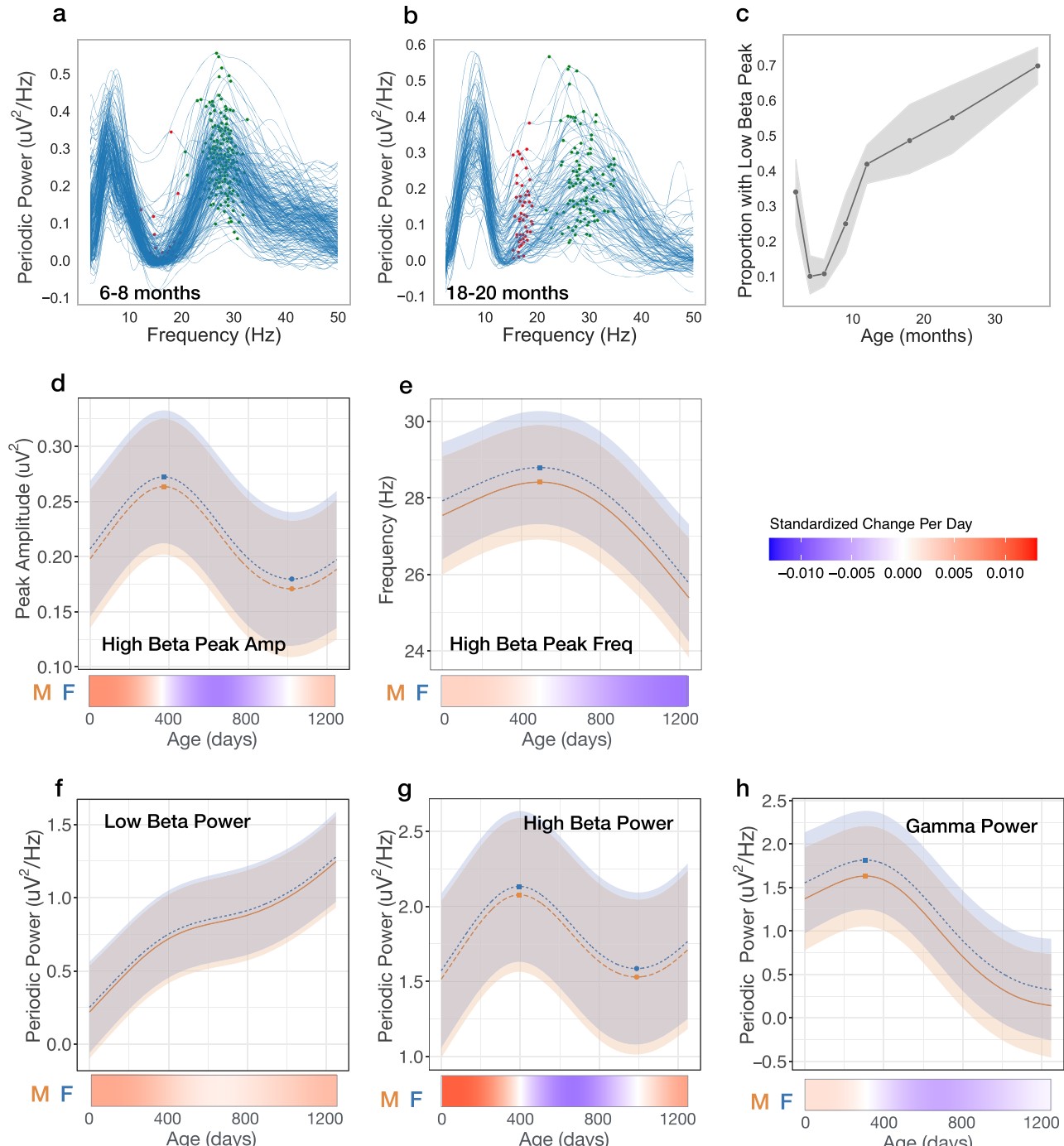

**Fig. 3 | Transient and nonlinear changes in periodic power between 12 and 35 Hz. a, b** Individual periodic power spectra 6–8 months, and 18–20 months old. Red markers show peaks between 12 and 20 Hz. Green markers show peaks between 20 and 35 Hz. **c** Proportion of infants with peak identified between 12 and 20 Hz at each age bin, with shaded areas representing 95% confidence intervals. **d–h** GAMMS modeled trajectories for males (orange) and females (blue). Lines are the model predicted value with the shaded area representing 95% confidence intervals. Relative inflection points are shown with circular markers. Source data are provided as a Source Data file. Below, heatmaps show the standardized change in offset or slope per day, defined as [change per day]/[standard deviation of measures across full age range].

adulthood. First, we observe increases in both aperiodic offset and slope, especially during the first year, whereas decreases in both measures are observed starting as early as 4 years of age and continue to decrease with adulthood[13–16]. Second, while expected shifts in the dominant peak from the theta to alpha range were observed between 5 and 44 months, in the 2–4 months age bin, a 9.5 Hz peak was also transiently observed. Third, striking changes within the beta (12–30 Hz) range were observed, including the emergence of a low

beta peak starting after 6 months of age, and age-dependent shifts in high beta peak frequency and amplitude - first increasing and then decreasing with age. Below we discuss how the above age-related changes may represent sequential developmental maturation in the inhibitory system and thalamocortical network connections.

The aperiodic offset is hypothesized to represent broad band neuronal firing[10,47], and thus early increases in aperiodic offset are consistent with established increases in neuronal number, gray matter

a

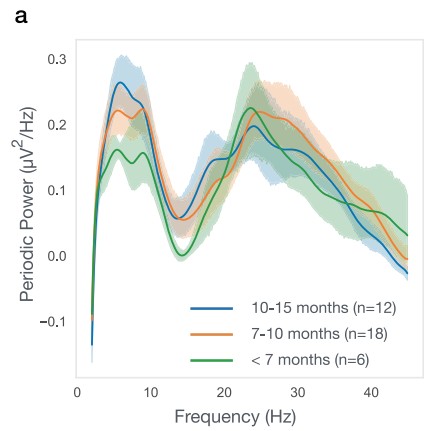

b

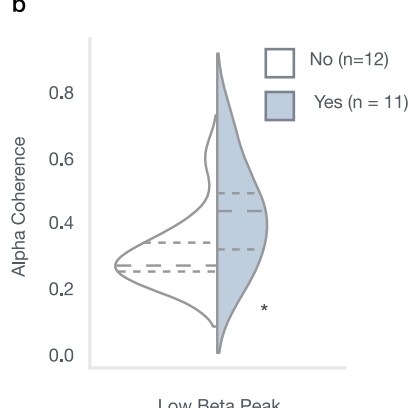

**Fig. 4 | Increased anesthesia-induced alpha coherence in infants with identified low beta peak in baseline EEG. a** Periodic power spectra of infants between 6 and 15 months old prior to receiving anesthesia. Mean spectra shown with shaded errors describing 95% confidence intervals. **b** Mean alpha coherence during anesthesia in infants 7–12 months old, with (light blue) or without (white) an identified low beta peak. Both a generalized linear mixed effects (glme) model and Ancova showed significant association between low beta peak presence with anesthesia-induced alpha coherence ($p = 0.019$ glme; $p = 0.03$ Ancova). Dotted lines represent quartiles, with the long dash represented the median.

volume, and synaptic number during the first year[1–3,5,6,8,48–50]. Stabilization of aperiodic offset after 1 year of age is also consistent with MRI findings that gray matter volume doubles during the first postnatal year and then slows to 20% in its second year[48,49,51]. Regionally, we also observe differences between posterior and frontal aperiodic offset trajectories, which either plateau after 1 year (posterior), or have a slow continued increase (frontal) beyond 1 year of age (Supplemental Fig. 3). Consistent with this pattern, synaptogenesis differs across cortical regions, with the posterior visual cortex exhibiting a burst in synapse formation between 3 and 4 months of age, whereas the prefrontal cortex shows peak synaptogenesis around 8 months of age and continued gains during the second year of life[1].

Our observed age-dependent increases in aperiodic slope in infancy also contrast with multiple studies covering child to adulthood, where decreases in slope have consistently been reported. Schaworonkov et al.[52] also reported decreased slope with age in infants from 1 to 7-months-old, however the parameterization of the spectra in that study was limited to 1-10 Hz due to excessive muscle noise in the data, and it is unclear how the shifts in 4-12 Hz periodic power described below may affect modeling of the underlying aperiodic component in this range. We hypothesize that observed increases in aperiodic slope reflect changes in inhibitory networks that are unique to early development. Indeed, aperiodic slope from EEG recorded from sleeping newborns is observed to increase with age during the first 7 weeks after birth[53]. Growing evidence suggests that aperiodic slope is modulated by the balance between excitation and inhibition, with increased slope associated with a reduction in E/I ratio[11,14,50]. An age-dependent reduction in E/I ratio during the first postnatal year is consistent with the prolonged developmental timing of inhibitory network maturation in humans. Unlike excitatory neurons which are well established by birth, during the first postnatal year GABAergic inhibitory neurons continue to migrate from ventral subregions of the brain to the cortices where they ultimately mature and integrate into neuronal networks[54]. In addition, during this 1st year GABAergic responses switch from being excitatory to inhibitory due to changes in the concentration of chloride channels on cell membranes[55–57]. Overall, inhibitory neuron network integration and the excitatory-to-inhibitory GABA switch are unique to this developmental period and likely lead to increased inhibitory tone during the first year.

Observed changes in the periodic spectra may reflect sequential steps in inhibitory network and thalamocortical circuit development. Transient neural circuits are common in postnatal development and play critical steps in normal development of thalamocortical circuitry[6].

For example, transient circuits between sublate neurons (SPN) and thalamo-recipient layer 4 spiny stellate neurons help establish thalamocortical connections prior to the maturation of primary sensory cortices[58]. Studies of postmortem fetal monkey and human brains suggests that the SPN in primates and humans slowly begins to disappear in the 3rd trimester but may persist until 6 months, with an overlapping period in which the thalamus makes connections with both the SPN and cortical layer IV neurons[6,59,60]. We hypothesize that the 9.5 Hz peak observed at 2–4 months, but not at 6 months, reflects this transient period when mature excitatory subplate neurons are still receiving and relaying thalamic input to cortices, resulting in higher frequency alpha oscillations. Additionally, newly established connections between the thalamus and layer IV produce lower frequency theta rhythms that will later become the dominant alpha rhythm.

The thalamus is thought to play a central role in the generation of the mature posterior alpha rhythm[61–63]. A shift in dominant oscillatory frequency in the theta/alpha range (4–12 Hz) across early childhood has been observed now in many studies[28,32,64]. Here, we both confirm and extend those findings over the first 3 years after birth, with peak frequency increasing most between 4 and 18 months. What factors are potential contributors to this shift in peak frequency? The dynamic circuit motif model (DCM) proposes that cortical network rhythms result from a combination of the intrinsic resonant frequency of a neuronal population and the time course properties of the inhibitory inputs on the neuronal population[65]. Under the DCM model, prior to the maturation of both local inhibitory circuitry and thalamocortical feedback loops, peak frequency oscillations as measured by scalp EEG are more likely to represent the intrinsic properties of cortical networks, with thalamic inputs beginning to play a larger role with age. For example, lower frequency 4–7 Hz oscillations are intrinsically generated by isolated layer 5 cortical neurons, and the range of oscillations increases to 5–12 Hz when connections to other cortical layers remain intact[66]. Thalamic neurons in the lateral geniculate nucleus also fire across the theta and alpha range. In vitro slice experiments from cats suggest that cortical input to thalamus modulates whether theta versus alpha oscillations are dominant[22,23]. Thus, the developmental shift in peak frequency from the canonical theta to alpha range over the first three years after birth, may represent the integration and maturation of cortical inhibitory neurons, as well as the establishment and maturation of thalamocortical connections.

Finally, our study identified early age-dependent changes in periodic beta power that we hypothesize are associated with thalamocortical loop maturation. We observe the emergence of a low beta

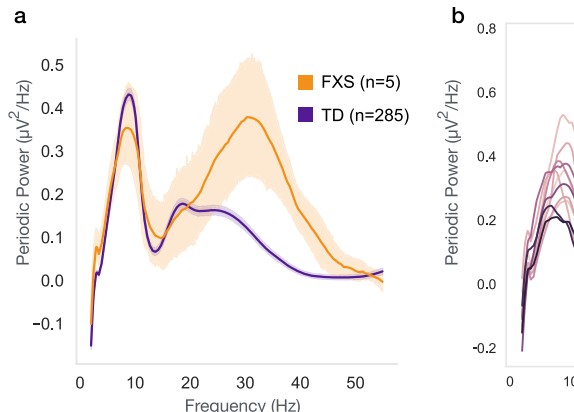
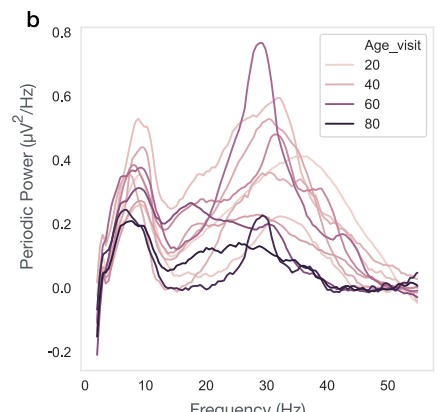

**Fig. 5 | Children with Fragile X Syndrome (FXS) have increased high beta peak that decreases with age. a** Periodic power spectra of 35–48 month-old children with and without FXS. Mean spectra shown with shaded errors describing 95% confidence intervals. **b** Individual periodic power spectra of FXS children from 32–84 months old, with line hue corresponding to age in months.

peak in infants older than 6 months of age and find that the presence of a low beta peak is associated with higher anesthesia-induced frontal alpha coherence. Biophysical models demonstrate that this frontal anesthesia-induced alpha coherence requires inputs from both the thalamus and cortex[38]. Together these findings suggest that low beta oscillations may directly reflect thalamocortical loop maturation. Beta rhythms are thought to be generated both locally in the cortex through pyramidal-interneuron loops, as well as through thalamus to cortical connections that also rely on inhibitory inputs[33]. The emergence of the low beta peak in awake infants may reflect the combination of newly established network connections between thalamic nuclei and cortical layers, as well as the maturation of interneurons within the thalamo-cortical pathways.

It is also possible that developmental changes in beta power are related to infant movement. During EEG acquisition, infants are held in their parent's lap and behavioral supports are in place to keep the infants calm. However, it is not possible to control the infants' move-ment, and movements both small (hand movements) and large (head turns, leg and arm movements) ubiquitously occur across recordings - likely increasing over the first year as infants become more mobile. Our preprocessing artifact removal pipelines (see Methods) includes sev-eral steps for removing high frequency noise from muscle artifact. However, this would not remove EEG signal in response to sensor-imotor processing. Infant jaw and upper limb movements have been shown to increase power between 9 and 20 Hz along frontal and occipital sites, while hand and lower limb movements do not have significant effects[67]. In our dataset, increases in low beta power were most prominent in central (not frontal or occipital) ROIs (Supple-mental Fig. 3K). In addition, age-dependent shifts of both low and high beta are observed visually across individual power spectra plots (Supplemental Fig. 2) and similar age-dependent shifts are observed in infants who were noted to have either limited or substantial movement during baseline EEG acquisition (Supplemental Fig. 9).

Transient high beta peaks were also observed across this early period of development. Specifically, we observed early increases in high beta peak amplitude, which reached a maximum at 12.2 months, followed by decreases in both high beta peak amplitude and fre-quency, such that by 36 months the low beta peak is the dominant peak across the 12-30 Hz range. The neurobiological mechanism of this high beta peak is unclear. As discussed above, administration of GABA agonists induces beta activity. In addition, several neurodevelop-mental disorders associated with GABA receptor dysfunction show prominent beta peaks on EEG; individuals with Duplication 15q have a prominent beta peak at 23Hz[68,69], and we have observed that children with Fragile X Syndrome (FXS), aged 3–7 years, have a prominent

30 Hz peak[70]. This 30hz peak observed in FXS children is qualitatively similar to the 30 Hz peak observed in the present dataset at a much younger age. Further analysis of data previously published from FXS children shows that the observed high beta peak decreases with age (Fig. 5), suggesting delayed brain development. Such observations highlight the value of the longitudinal EEG trajectories presented in this paper in placing findings from neurodevelopmental disorders in the broader context of developmental brain maturation.

There are several limitations that may affect the interpretation or generalizability of findings. First, there are significant non-neural ana-tomical developmental changes that occur during infancy, including increases in skull thickness, gradual closure of fontanels and sutures, and changes in cerebrospinal fluid volume. Such changes can alter conductive properties of the skull, and in turn impact EEG signals, including measures such as aperiodic offset and slope[71–73]. Second, more work is needed to firmly establish the association between changes in periodic power and development of thalamocortical cir-cuitry. While we provide some preliminary evidence that the emer-gence of low beta peak is associated with maturation of thalamocortical connectivity, the dataset is small and it is still an indirect measure, and thus no conclusion can be made at this time. Future research that combines EEG and MRI in infants from 4–6 months and again at 10–15 months, could provide more direct evi-dence linking alpha and beta periodic EEG findings with the develop-ment of thalamocortical connectivity. Third, impedance thresholds were kept below 100 kΩ, rather than 50 kΩ, in order to reduce the time needed to optimize impendences and in turn prevent infants and toddlers from becoming fussy. While all data were collected in elec-trically shielded rooms, it is possible that recordings with impedances above 50 kΩ have reduced signal quality. However the developmental changes described are robust and observed across individuals as observed in Supplemental Fig. 2. Fourth, the findings presented here utilize a specific artifact removal and processing pipeline (BEAPP and HAPPE1.0). An automated pipeline was used to improve reproduci-bility of our findings, and the code is available on Open Science Fra-mework (https://osf.io/u3gp4). We do note that similar developmental changes in the low and high beta from 9 and 12 months are shown in Fig. 1 of Rayson et al.[74] which used a combination of MADE[75] and NEAR[76] pipelines. Interestingly, recently Rico-Pico et al.[31] also described developmental changes in the periodic power spectra using MADE processing pipelines but limited their analysis to 1–20 Hz "due to a power bump" in the gamma range, assumed to be related to muscle artifact, but likely the same high beta peak we observe most prominent at 7–8 months of age. Power spectra from 6, 9, and 16 months shown in Supplemental Fig. 8 of Rico-Pico et al. 2023 visually show a similar

pattern in beta band change across development. We hope that our comprehensive characterization of developmental changes using a single pipeline with frequent age sampling across the first 3 years of development spur other infant EEG researchers to evaluate the patterns we describe in their own data.

In summary, our work highlights the dynamic developmental changes in neural activity occurring during the first three years after birth and provides insights in potential ways these age-dependent and sometimes transient changes may coincide with sequences in thalamocortical and inhibitory network maturation. Our findings help to ground cross-sectional work occurring at these early ages and provide a foundation to compare developmental trajectories of various neurodevelopmental disorders including autism, ADHD, and rare genetic disorders. Future studies examining early trajectories of functional connectivity and phase amplitude coupling across this age range will provide additional insight into the timing of critical periods in brain maturation.

## Methods

### Studies and participants

Lab-based EEGs for this paper were collected as part of four different studies occurring over 15 years conducted at our lab at Boston Children's Hospital (Fig. 1A). Institutional review board (IRB) approval was obtained from Boston Children's Hospital. IRB protocol numbers are provided for each study. Written, informed consent was obtained from a parent or guardian prior to each infant or child's participation in the study. Sample numbers for each age are shown in Table 1. Study 1, the Healthy Baby Study (IRB-P00019083), was a longitudinal study, enrolling infants starting at 2 months of age, from the Boston Children's Hospital Primary Care Center, which predominantly services families from low-income backgrounds. EEG was collected and developmental assessment using the Mullen Scales of Early Learning (MSEL) was performed at 2, 6, 9, 12, 24, and 36 months.

Study 2, The Infant Sibling Study (IRB-X06-08-0374), and Study 4, the Infant Screening Project (IRB-P00018377), were both prospective, longitudinal studies, enrolling infants with and without first degree family history of ASD starting as early as 3-months of age. For this analysis only infants without family history of ASD were included. Study 4 also included a group of infants with elevated social communication concerns at 12 months of age, and they were also excluded from this analysis. EEGs and MSEL were performed at 3, 12, 18, 24, and 36 months for both studies, as well as 6 and 9 months for Study 2. Infants were specifically assessed for ASD using the Autism Diagnostic Observation Schedule (ADOS) in conjunction with clinical best estimate at 24- and 36-month visits.

Study 3, the Emotion Project (IRB-P00002876), was a cohort/longitudinal study. Infants were enrolled at either 5, 7, or 12 months, and then followed through 7 years of age. In addition to the first time point, EEG data was again collected at 3 years of age. There were no developmental assessments performed for this study, however parent questionnaires regarding child development, diagnoses (e.g., ASD), and therapies were collected.

All infants had a minimal gestational age of 36 weeks, no history of prenatal or postnatal medical or neurological problems, and no known genetic disorders. Infants who were later diagnosed with ASD (either by assessment during the study, or by community diagnosis disclosed by parents prior to age of 5) were not included in this study.

Sample characteristics across and within studies are shown in Table 1. The analysis included 1335 EEGs collected from 592 participants. While all studies took place in the same laboratory, participant demographics vary between studies as was expected given differences in recruitment and research aims of each study. In addition, studies differed in the age of enrollment and when subsequent visits were completed (Fig. 1C). Combined, the sample remained predominantly white (74%).

### Lab-based EEG data collection

Baseline, non-task-related EEG data was collected using similar methods and rooms for all four studies. The infant was held by their seated caregiver in a dimly lit, sound-attenuated room with a low-electrical-signal background. For Study 2, a research assistant ensured that the infant remained calm by blowing bubbles and/or showing toys. For Studies 1 and 3, a video of infant toys was shown for 2−5 minutes and 2 min, respectively. For Study 4, a video of an abstract moving objects was shown for 2−5 minutes. Continuous scalp EEG for Studies 1, 3, and 4 was recorded using a 128-channel Hydrocel Geodesic Sensor Nets (Electrical Geodesics, Inc., Eugene, OR) connected to a NetAmps 300 amplifier (Electrical Geodesic Inc.) and sampled at 500 Hz. Study 2 included recordings using 64-channel Geodesic Sensor (<10% of data) or a 128-channel Hydrocel Geodesic Sensor Nets (Electrical Geodesics, Inc., Eugene, OR), connected to either a NetAmps 200 or 300 amplifier (Electrical Geodesic Inc.) and sampled at either 250 or 500 Hz. Additional statistical analysis related to differences in net and amps is described below in EEG Power analysis. For all studies, data was referenced online to a single vertex electrode (Cz) and impedances were kept below 100kΩ in accordance with the impedance capabilities of the high-impedance amplifiers inside the electrically shielded room. Electrooculographic electrodes were removed to improve the child's comfort.

### EEG pre-processing

Raw Netstation (Electrical Geodesics, Inc) files were exported to MATLAB (version R2017a) for preprocessing and absolute power calculations using the Batch Automated Processing Platform (BEAPP[77]) with integrated Harvard Automated Preprocessing Pipeline for EEG (HAPPE[78]). For each EEG, a 1 Hz high-pass and 100 Hz low-pass filter were applied, data sampled at 500 Hz were resampled to 250 Hz, and then run through the HAPPE module consisting of 60 Hz line noise removal, bad channel rejection, and artifact removal using combined wavelet-enhanced independent component analysis (ICA) and Multiple Artifact Rejection Algorithm (MARA[5,6]). The following channels, in addition to the 10−20 electrodes, were used for MARA: 64-channel net − 16, 9, 8, 3, 58, 57, 21, 25, 18, 30, 43, 50, 53, 32, 33, 38, 41, 45; and 128-channel net - 28, 19, 4, 117, 13, 112, 41, 47, 37, 55, 87, 103, 98, 65, 67, 77, 90, 75. These electrodes were chosen as they evenly cover all brain regions of interest (Supplemental Fig. 1). After artifact removal, channels removed during bad channel rejection were then interpolated, data were referenced to the average reference, detrended to the signal mean, and segmented into 2-second segments. Any segments with retained artifact were rejected using HAPPE's amplitude and joint probability criteria.

### EEG rejection criteria

EEG recordings were rejected using the following HAPPE data quality measures: Fewer than 20 segments (40 s of total EEG), percent good channels <80%, percent independent components rejected >80%, mean artifact probability of components kept >0.3, and percent variance retained <25%. Expected differences between studies in number of segments remaining post-segment rejection were observed, with Study 3 with the shortest resting state recording period having fewer segments. All other quality metrics were similar across studies (Table 2).

### EEG power analysis

The power spectral density at each electrode, for each 2-second segment, was calculated in the BEAPP Power Spectral Density (PSD) module using a multitaper spectral analysis[44] and three orthogonal tapers. For each electrode, the PSD was averaged across segments, and then further averaged across all available electrodes, or frontal, temporal, central, and posterior regions of interest (Supplemental Fig. 1C, D). The PSD was then further analyzed using a modified version

**Table 2 | EEG data quality metrics**

|  | Combined Studies $N = 1335$ | Study 1 $N = 206$ | Study 2 $N = 314$ | Study 3 $N = 519$ | Study 4 $N = 296$ |
|---|---|---|---|---|---|
| **EEG quality metrics, mean ± SD** |  |  |  |  |  |
| Number of segments retained after processing | 81 ± 39.5 | 126 ± 27.5 | 85.8 ± 39.9 | 48.8 ± 7.1 | 105.2 ± 30.3 |
| Percent good channels | 92.2 ± 4.5 | 92.2 ± 4.8 | 92.6 ± 4.4 | 92.0 ± 4.6 | 93.5 ± 4.3 |
| Percent ICs rejected | 35.7 ± 10.3 | 35.7 ± 10.6 | 35.4 ± 10.1 | 34.1 ± 14.0 | 35.0 ± 10.5 |
| Percent variance kept of post waveleted data | 67.2 ± 17.0 | 63.1 ± 16.1 | 66.5 ± 15.7 | 68.8 ± 18.3 | 68.1 ± 16.1 |
| Mean artifact probability of kept ICs. | 0.12 ± 0.05 | 0.12 ± 0.04 | 0.11 ± 0.04 | 0.12 ± 0.05 | 0.12 ± 0.05 |

of SpecParam v1.0.0[8] (https://github.com/fooof-tools/fooof; in Python v3.6.8) in order to model periodic and aperiodic components of the power spectra. SpecParam required modification for use in this age range, as power spectrum models for 2–7 month ages showed poor model fit (increased mean squared error) for frequencies between 10 and 20 Hz. Specifically, the SpecParam modeled curves did not accurately capture the "trough" in the power spectra visually observed in this frequency range at younger ages (Supplemental Fig. 7 and 8) when alpha and high beta peaks are prominent, but a low beta peak is not present. In the original SpecParam, the power spectrum is first fit with an estimated aperiodic component which is subtracted from the raw signal. Any negative data, signal falling below 0, is converted to 0, and peaks are identified through iterative gaussian fits. Once all peaks are identified, these peaks are removed from the raw spectra and a final aperiodic component is fit. Between 2 and 7 months, signal between 10 and 20 Hz often falls below the first estimated aperiodic fit and is thus converted to 0, impacting both the periodic peaks identified and the final fit of the aperiodic component. To improve model fit, the robust_ap_fit function, which initially defines the aperiodic component, was modified so that the initial estimate of the flattened power spectra (flatspec) has a baseline elevated such that the lowest point is ≥ 0, to avoid omitting data important across the 2–7 month age range. In the original and modified scripts, this initial fit is combined with thresholding to render a more robust second round of aperiodic parameters. After these second aperiodic parameters have been defined, the fit function re-estimates the flattened spectra (spectrum_flat). At this point, prior to fitting spectra peaks, the modified code sets negative data in the flattened spectra equal to 0, similar to the approach of the original code during the initial aperiodic fit. In both versions, aperiodic parameters are fit a third and final time to the spectra with peaks removed (spectrum_peak_rm). The SpecParam model was used in the fixed mode (no spectral knee) with peak_width_limits set to [0.5, 18.0], max_n_peaks = 7, and peak_threshold = 2. Code is available (osf.io/u3gp4) which runs both the original and modified versions of SpecParam (with edits marked in code). Changes are also shown in Supplemental Materials. The code also graphs the RMSE across frequencies for both versions, separated by age. Comparisons model fits for each age for are presented in Supplemental Fig. 4, as well as RMSE across frequencies for infants 5, 6, or 7 months old. Further analyses were subsequently restricted to 2.5-50 Hz given elevated error between 2 and 2.5, and 50 and 55 Hz. Mean $R^2$ for the full sample using this modified version of SpecParam was 0.997 (STD 0.008; range 0.890-0.999). Mean $R^2$ for each age bin ranged from 0.991-0.999. The mean estimated error for the sample was 0.01 (STD 0.01, range 0.002–0.09).

SpecParam provides two parameters to describe the aperiodic 1/f signal: offset and slope. As the SpecParam-determined offset is extrapolated to the estimated aperiodic power at 0 Hz, where there are high amounts of error, we instead calculated the aperiodic offset based on aperiodic power at 2.5hz (Fig. 1a). The periodic power spectrum (Fig. 1b, f) was determined by subtracting the SpecParam estimated aperiodic spectrum (Fig. 1e) from the absolute power spectrum (Fig. 1d). To further characterize peaks within the power spectra across

development, the periodic spectrum was then smoothed using a savgol filter (scipy.signal.savgoal_filter, window length = 101, polyorder = 8). Individual periodic power spectrum plots before and after savgol filter are shown in Supplemental Fig. 2. We decided to use this method instead of using the SpecParam estimated peak_fit as the high beta peak appeared to have a non-gaussian shape at some ages, and thus peak_fit estimates did not accurately identify the high beta peak frequency. Using the smoothed periodic spectra, maxima were identified within the following frequency ranges: 4–6.5 (theta), 4–12 Hz (theta/alpha), 12–20 Hz (low beta), and 20–35 Hz (high beta). Aperiodic and periodic power across the following canonical frequency bands was calculated taking the integral of each parametrized spectra between the following frequency ranges: theta (4–6 Hz), low alpha (6–9 Hz), high alpha (9–12 Hz), low beta (12–20 Hz), high beta (20–30 Hz), and gamma (30–45 Hz).

To visualize differences aperiodic and periodic EEG measures across different regions, topoplots for 12 measures across eight age bins are shown in Supplemental Fig. 6. As SpecParam estimates were based on power spectra across averaged electrodes, topoplots were similarly created using averaged measures across electrodes from 6 regions of interested (left frontal, right frontal, central, left temporal, right temporal, and posterior).

As Study 2 collected data with 2 net types and 2 amplifiers, data from 6-, 9-, and 12-month age bins were assessed for spectra differences in total (2-50hz) aperiodic and periodic power as well as aperiodic slope and intercept measure from central electrodes between either 64 and 128 channel nets, or NetAmps 200 or 300 amplifiers. Of the 24 analyses performed, 3 showed significant differences. Net-type differences were observed for 9- and 12-month central aperiodic slope ($p = 0.04$ for both) and an amplifier difference was observed for 12-month central offset ($p = 0.04$). None of these were significant after correcting for multiple comparisons.

**Anesthesia cohort**
EEG data were collected from infants undergoing anesthesia as part of a prospective observational study approved by the Institutional Review Board at Montefiore Medical Center, Albert Einstein College of Medicine[46,79]. Infants scheduled for elective surgical procedures (e.g., circumcision, hernia repair) were recruited. Infants were excluded for prematurity, known neurologic injury, epilepsy, or planned intracranial surgery. Infants less than 6 months of age, or those documented to be asleep or crying during baseline (pre-anesthesia) recordings were excluded from further analysis. All subjects received general anesthesia with sevoflurane.

EEG recordings were obtained using a Food and Drug Administration (FDA) approved 26 channel device recording from scalp locations designated by the International 10-20 System, reference midline occipital channel (Oz) (microEEG System, Biosignal, Acton, MA)[80]. Data was collected from 21 electrodes (FpZ, Fp1, Fp2, Fz, F3, F4, F7, F8, Cz, C3, C4, T3, T4, Pz, P3, P4, T5, T6, Oz, O1, O2) both prior to (baseline) and during sevoflurane induction and maintenance. End tidal sevoflurane concentration was recorded, locked in time with EEG recording. Data were sampled at a frequency of 250 Hz.

Baseline EEGs ($n = 45$) were visually inspected and approximately 2 min of continuous EEG with minimal artifact was segmented and then processed using BEAPP/HAPPE[77,78]. 9 EEGs were excluded due to excessive artifact during visual inspection. No additional EEGs were excluded based on HAPPE data quality criteria. The final sample ($n = 36$) had an average age of 9.1 months (range 6–15 months), and was predominantly male ($n = 25$). Central periodic power using Pz and Fz electrodes was calculated using multitaper spectral analysis using three orthogonal tapers. PSD was then further analyzed using a modified version of SpecParam v1.0.0 as described above, in order to model periodic and aperiodic components of the power spectra. A low beta peak was identified as described above from the baseline EEG.

### Alpha coherence analysis

Preprocessing of Anesthesia EEG data: We used a bipolar montage to analyze the data (F7-Fp1 and F8-Fp2). We developed an automatized method to exclude epochs with high-amplitude noise based on the standard deviation of the time series signal. Three members of the team (CW, JC, and RG) visually inspected the remaining epochs to select all available 30-second artifact-free segments. We then selected all epochs with a stable sevoflurane concentration defined as two consecutive minutes of end-tidal sevoflurane levels within 0.2% preceding the selection of an epoch of data. EEG data were band-pass filtered from 0.1 to 30 Hz.

For each subject, corresponding EEG data collected during anesthesia were inspected to identify all 30 s segments where both the sevoflurane concentration was stable, and segments were "artifact free" (eg. no motion or electrocautery artifacts). Epochs with high-amplitude noise in frontal electrodes (F7-Fp1; F8-Fp2) based on standard deviation of the time series were automatically excluded. Blinded visual inspection by members of the team (CW, JC, and RG) identified remaining epochs with stable sevoflurane concentration without other anesthesia (e.g. propofol bolus) interference. Stable sevoflurane concentration was defined as two consecutive minutes of end-tidal sevoflurane levels within 0.2% preceding the selection of an epoch of data.

Coherence Analysis: EEG data were band-pass filtered from 0.1 to 30 Hz. Custom EEG analysis scripts were written using MATLAB (version R2021a, MathWorks, Natick, MA), employing functions in the Chronux toolbox[81]. Coherence analysis was calculated between F7 and F8 using the multitaper method with the following parameters: window length $T = 6$ s with no overlap, time- bandwidth product TW = 3, number of tapers $K = 5$, and spectral resolution of 2 W = 1 Hz. We calculated coherence by quantifying the degree of correlation between both signals across a frequency range as previously described[82]:

$$C_{xy}(f) = \frac{\left|S_{xy}(f)\right|}{\sqrt{S_{xx}(f)S_{yy}(f)}}$$

(1)

Where $S_{xy}(f)$ is the cross-spectrum between the signals $x(t)$ and $y(t)$ (i.e., F7 and F8 electrodes), $S_{xx}(f)$ is the power spectrum of the signal $x(t)$ and $S_{yy}(f)$ is the power spectrum of the signal $y(t)$. Then, the median coherence within the alpha band was used for analyses.

### Statistical analyses

GAMMs: To assess developmental trajectories of power spectral measures, we used generalized additive mixed models (GAMM). GAMMs are similar to generalized linear mixed models, with the advantage that predicts can be modeled linearly and nonlinearly. In GAMMs, smooth linear or nonlinear functions of the relationship between predictors and the outcome are simultaneously estimated, and then summed. GAMM is therefore an advantageous framework for exploring the relationship between power spectral measures and age, for which the underlying form of the relationship is not yet known.

Models were fit using *mgcv* package[83] (version 1.8-38) and R (version 4.1.2).

A separate GAMM was fit to predict each power spectral measure, for each region of interest (e.g., whole brain theta power, frontal aperiodic offset). First, to determine whether to include an age-by-sex interaction, two models were fit with the following forms:

$$\text{Power Measure} \sim \text{oSex} + s(\text{age\_days}, k = 4, fx = T) + s(\text{New\_ID}, bs = 're') + s(\text{Study}, bs = 're')$$

(2)

$$\text{Power Measure} \sim \text{oSex} + s(\text{age\_days}, k = 4, fx = T) + s(\text{age\_days}, by = \text{oSex}, k = 4, fx = T) + s(\text{New\_ID}, bs = 're') + s(\text{Study}, bs = 're')$$

(3)

oSex represents sex stored as an ordered factor; coding sex as an ordered factor is necessary for the GAMMs model to produce a single significance value for the age-by-sex interaction. This is more interpretable than the alternative of including sex as a categorical factor where two separate smooth effects of age are modeled for each sex, but no direct comparison is provided. s(age_days, $k = 4$, fx = T) is a smoothed age term. Study and New_ID are each included as random effects to account for repeated observations and clustering of observations within studies. These models were compared by ANOVA, and model 2 (including the interaction term) was chosen if the difference was significant ($p < 0.05$). To correct for multiple comparisons, the false discovery rate (FDR) was controlled using the Benjamini and Hochberg method[84], which was applied for each model term within each region of interest across the 16 measure types (e.g., theta power, beta power, aperiodic offset) to produce q-values (FDR-corrected $p$-values).

To further understand the nonlinear trajectories of change, inflection points were calculated using the argrelextrema function from scipy in python with order = 100. A standardized rate of change per day was calculated to visualize developmental changes within features. The modeled value of a feature at a given age (in days) was subtracted from the modeled value from the subsequent day, and this was divided by the standard deviation of the modeled values of that feature across the age range.

To assess the differences between regions of interest, GAMM models were fit with the following form:

$$\text{Power Measure} \sim s(\text{age\_days}) + \text{oSex} + \text{ROI} + s(\text{New\_ID}, bs = 're') + s(\text{Study}, bs = 're')$$

(4)

where the terms have the same meanings as above, and ROI is a factor representing the four regions (frontal, central, temporal, posterior). Because prior literature and preliminary visual inspection of the data indicated that the posterior ROI is most unique in the time course of development, the posterior ROI was set as the reference factor. Thus, the effect and significance associated with each of the other ROIs is a measure of the difference between that ROI and the posterior ROI.

To assess the relationship between age and probability of having one or two peaks within the broad alpha range at 2–6 months, a generalized linear mixed effects model was evaluated using the lme4 package in R[85].

Anesthesia Statistical Analysis: ANCOVA, with sevoflurane levels as a covariate, was used to determine effects of presence of low beta peak on anesthesia-induced alpha coherence.

Figures were created using Python v3.6.8 and python data visualization libraries *matplotlib* (https://matplotlib.org/) and *Seaborn* (https://seaborn.pydata.org/index.html) or in R (version 4.1.2).

### Reporting summary
Further information on research design is available in the Nature Portfolio Reporting Summary linked to this article.

## Data availability

Consents obtained from human participants at our institution prohibit sharing of identifiable and de-identified individual data without a data use agreement in place. Please contact the corresponding author with data requests. Source data are provided with this paper for Figs. 1g-i, 2e-i, and 3d–h and Supplemental Figs. 5 and 6. Source data are provided with this paper.

## Code availability

Code used for EEG processing and analyses used in this paper can be found on the Open Science Framework (https://osf.io/u3gp4).

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

## Acknowledgements

We thank all the children and families who generously participated in this research. We thank all the research staff involved in participant recruitment, data collection, and database administration. Funding Support: This research was supported by the National Institutes of Health (R01-DC010290 and MH078829 to C.A.N., K23DC07983, and T32MH112510 to C.L.W., and UL1TR002556, KL2TR002558, and K23DA057499 to J.Y.C.). The research was also supported by the JPB Research Network on Toxic Stress to CAN.

## Author contributions

Carol Wilkinson: conceptualization, methodology, software, formal analysis, data curation, writing – original draft, and visualization. Lisa Yankowitz: methdology, software, formal analysis, writing – review & editing, and visualization. Jerry Y. Chao: data collection (anesthesia cohort), methodology, formal analysis, investigation, data curation, resources, and writing – review & editing. Rodrigo Gutiérrez: methodology, software, formal analysis, data curation, and writing – review & editing. Jeff L. Rhoades: software and writing – review & editing. Shlomo Shinnar: project administration, supervision, and writing – review & editing. Patrick L. Purdon: methodology, supervision, and writing – review & editing. Charles A. Nelson: investigation, resources, writing – review & editing, supervision, project administration, and funding acquisition.

## Competing interests

Dr. Patrick Purdon is an inventor on pending patents on brain monitoring using the electroencephalogram. One of these patents is under non-exclusive license by Massachusetts General Hospital to Masimo Corporation. Dr. Purdon receives institutionally-distributed licensing royalties for this license. Dr. Purdon is also a co-founder of PASCALL Systems, Inc., a start-up company developing closed-loop physiological control for anesthesiology. The remaining authors have no competing interests.
