## [Peer Review File · Nature Communications]

Developmental trajectories of EEG aperiodic and periodic components in children 2-to-44 months of age.Reviewer #1 (Remarks to the Author):

This manuscript by Wilkinson and colleagues is a good addition to the growing field of developmental studies examining the development of EEG aperiodic and periodic activity. They demonstrate non-linear changes in aperiodic and periodic power features in a large developmental dataset of resting-state EEG from 592 infants aged 2–44 months. The authors found changes in aperiodic offset and aperiodic slope, as well as periodic alpha, low beta, high beta, and gamma power. They link these aperiodic slope and alpha and beta power changes to the developmental progression of thalamocortical circuitry, framing these findings in the established literature and demonstrating the power of investigating these electrophysiological changes as a window to the underlying physiology. They also present evidence that the presence of low beta is associated with thalamocortical connectivity sufficient for anesthesia-induced alpha coherence.

While the manuscript is overall interesting, we have a number of comments and suggestions, listed below.

* Periodic changes reflect underlying developmental changes in thalamocortical circuitry: The major argument of the paper is that changes in aperiodic slope and changes in oscillatory alpha and beta oscillations reflect changes in the development of the underlying thalamocortical circuitry. There is thorough evaluation of the current literature and rationale for how this could be the case, and the preliminary evidence that low beta is associated with thalamocortical connectivity sufficient for anesthesia-induced alpha coherence helps make the argument. But given it is the primary argument of the paper, it would be nice to have another dataset that could be used to validate the larger hypothesis, or to soften the language on this topic.

* Within-subjects analysis: The high-powered, between-subject analysis demonstrates how these periodic changes in alpha and beta oscillations are non-linear and consistent with development of thalamocortical circuitry. Given that the data is longitudinal, the results would be strengthened if they could be also demonstrated within-subjects.

* Page 3, lines 35-36: Sentence starting with “The majority of neural oscillations...” should have a citation. This is an important sentence for establishing how the aperiodic and periodic changes found reflect the underlying changes in thalamocortical circuitry, and should cite the first papers that established this link.

* Page 5, lines 7-8: While the necessary details are provided in the methods section, it would be nice to say how the spectra were calculated (i.e., multitapers).

* Figure 1A-B: It would be nice to see the full power spectrum first, then with the aperiodic and periodic components separated (as is the case in Figures 1D-F). Right now, it may make the reader think there is a way to separate the aperiodic and periodic signals through frequency decomposition instead of through spectral parameterization.

* Page 9, lines 4-5: The framing of a lack of a low beta peak as a “trough in low beta” seems unnecessary and confusing for two reasons. First, for there to be an apparent trough in low beta, there has to be peaks in alpha and high beta, but there can be a lack of a low beta peak regardless of alpha and high beta peaks. If oscillatory changes are meant to be independent (at least in principle), the terminology we use to describe them should not rely on how oscillations of different frequencies relate to each other. Second, a trough in low beta could also be caused by poor model fits where the aperiodic portion of the power spectrum is particularly noisy. For these reasons, I feel it would be better to avoid the trough terminology as a whole.

* Topographic plots of the different features would be nice to see, to get a sense of the rough spatial distributions.

* q-values are adjusted p-values, presumably FDR, but FDR details aren't given and this isn't really explicated.

* It is unclear if coherence is across the two frontal bipolar pairs. Is this the case?

* In addition, there are many ways of estimating coherence that focus on phase only, or just amplitude, or that include both. The authors need to include these details in the methods.

* There are significant non-neural changes in this age range as well, such as changes in skull thickness, the closure of the skull sutures, etc. that can affect the signals under investigation. These can't exactly be controlled for, but they need to be discussed as potential limitations.

* Why is sex included as an ordered factor, instead of categorical? In terms of the output of the model I'm not sure it makes a difference in the math of the regressions, but I'm curious.

Reviewer #2 (Remarks to the Author):

I appreciate the opportunity to review this manuscript. This study examined the longitudinal trajectories of periodic and aperiodic EEG power in a large, normative sample of infants. Results suggest non-linear increases in aperiodic offset and slope. Moreover, the manuscript examines the presence and changes in periodic alpha and beta, after accounting for the aperiodic slope. Changes in the presence (and absence) of alpha and beta peaks are interpreted as developmental changes in thalamocortical circuitry. This interpretation is further supported by evidence that that emergence of the low beta peak is associated with alpha coherence during anesthesia – an effect known to depend on thalamocortical connectivity. The study provides novel and important findings that add to our understanding on normative development of brain in infancy and toddlerhood by utilizing new ways to analyze EEG data. Moreover, the manuscript is well written and has several notable strengths such as utilizing a large group of infants followed longitudinally and a pharmacological manipulation. However, I have a couple questions and suggestions that I believe would improve the manuscript and its potential impact:

1. The interpretation of beta power being related to infant movement is an important one. I do not agree with the authors that beta power would only be observed in frontal or occipital sites because movement-related beta is often seen in central sites (over motor cortex), as observed by the authors. I have two suggestions. One is that recent findings suggest that movement-related beta also changes with age in what I believe is a similar pattern as found by the authors (Rayson et al., 2023). If this is correct, this should alleviate concerns about the developmental effects being due to differences in movement, but even when matching for movements, similar developmental patterns in brain activity are observed. Second, a more labor-intensive solution would be to code some of the videos at different ages and show that the effects of interest are not explained by differences in movement.
2. Combining multiple studies is a clear strength of the manuscript. However, it is not clear how the authors are accounting for the nesting of children within studies. I see that the authors are controlling for study as a fixed effect. However, is the co-dependence between observations between studies accounted for in any way in the GAMMs? I am not as familiar with GAMMs, but if it were an MLM, it could be accounted for by nesting the random effects at different levels (e.g., EEG assessments are nested within children, which are nested within studies). Please clarify if something like this was done by clearly outlining which fixed and random effects were estimated.
3. I believe the authors are using previous versions of the HAPPE software to clean their data. The old version is known to overcorrect during artifact rejection, potentially taking away aspects of the signal. Why did the authors decide to use this method over the new version of the method (e.g., version 3; Monachino et al., 2022)? I believe the new algorithms fix many of the concerns from the previous version. It would be reassuring if the authors reprocessed at least some of the data and observed similar results or high correlations between the values used in the manuscript and ones reprocessed with newer versions of the software.
4. Several statements are missing a citation. For example, "The majority of neural oscillations observed in the power spectrum are the direct result of inhibitory and thalamocortical network responses to sensory input." Or "The thalamus is thought to play a central role in the generation of the mature posterior alpha rhythm."
5. For Figure 1, it is unclear how aspects of the figure are related to the analyses or if they

represent real data. For example, the authors display the data in age bins, but as far as I understand, the analyses were conducted using continuous age. It is unclear if Panels A and B display simulated data as an example, one participant, one age group, or the average of all participants – please clarify these points in the figure legend.

6. Is it possible that the early alpha peak transitions to the low beta peak? I was not able to tell as 4-6 months was not displayed on Figure 2.

7. For the analyses with anesthesia, are the authors analyzing a difference in coherence between before and during anesthesia? Namely, is the outcome of the ANCOVA a change score? Or how is the baseline recording being used?

8. I appreciate the supplemental analyses by ROI, but it would be useful to provide topoplots for the effects of interest. Moreover, some of the labels for the figures in the supplement are missing (Figure S3A, S3B)

9. What do the dotted lines stand for in Figure 4B?

10. It is unclear how many infants had longitudinal and data, and how many assessments each infant had. Please provide more details on exactly how many participants had longitudinal data and for which ages. A suggestion would be to create an additional table.

11. For Table 2, please add number/percent of segments retained after preprocessing.

12. Minor: For Table 1, it is strange to say that infants have income. I would suggest saying families rather than infants.

13. Minor: The name of FOOOF has been updated to SpecParam. Please use the updated name.

14. Minor: Please include page numbers.

References:

Monachino, A. D., Lopez, K. L., Pierce, L. J., & Gabard-Durnam, L. J. (2022). The HAPPE plus Event-Related (HAPPE+ER) software: A standardized preprocessing pipeline for event-related potential analyses. *Developmental Cognitive Neuroscience*, 57, 101140. <https://doi.org/10.1016/j.dcn.2022.101140>

Rayson, H., Szul, M. J., El-Khoueiry, P., Debnath, R., Gautier-Martins, M., Ferrari, P. F., Fox, N., & Bonaiuto, J. J. (2023). Bursting with potential: How sensorimotor beta bursts develop from infancy to adulthood (p. 2023.05.09.539976). *bioRxiv*. <https://doi.org/10.1101/2023.05.09.539976>

Reviewer #3 (Remarks to the Author):

The study titled "Early Development of Neural Circuits in Infants Revealed Through EEG Analysis" presents an investigation into the development of aperiodic and periodic brain activity in infants. The strength of the current study is its large sample size. However, there seems to be a lack of substantial novelty for such a high-impact paper as *Nature Communications*. Studies on longitudinal changes of aperiodic and periodic activity have been previously published (e.g. Schaworonkow & Voytec, 2021, Rico-Pico et al., 2023). The actual novelty of the paper is the investigation of the relationship between low beta peak developmental emergence of anesthesia-induced frontal alpha coherence (within a small sample).

More importantly, there are several aspects (analytical procedures) of the study that warrant critical examination (see more details below). As infant data usually exhibit lower signal quality (due to excessive motion artifacts and limited cooperation), the data and preprocessing need to be evaluated carefully and additional control analyses and visualization should be provided to ensure the validity of the results and its interpretation.

EEG acquisition:

The site should be included as random effect in the analysis. Even if the same equipment was used (for 3 out of 4 studies), it is expected that large site effects are observable (e.g. due to distinct technical personnel).

impedances were kept below 100kΩ is even very high for a Geodesic NetAmps amplifier.

EEG preprocessing:

The MARA algorithm is not trained for infants data and thus inadequate choice in the current analysis pipeline. From personal experience, I know that the ICA MARA algorithm can remove insufficient amount of components or possibly also being too conservative (removing too many components), which might be the case here: on average ~35 components were removed, which is approximately half-of the components included in the ICA. Consider using ICLabel or do manual inspection of the components. The fact that Moreover, it is unclear how the additional electrodes (MARA: 64-channel net – 16, 9, 8, 3, 58, 57, 21, 25, 18, 22 30, 43, 50, 53, 32, 33, 38, 41, 45; and 128-channel net - 28, 19, 4, 117, 13, 112, 41, 47, 37, 55, 87, 103,23 98, 65, 67, 77, 90, 75.) were implemented in MARA.

FOOOF parametrization:

The description of the FOOOF parametrization is insufficient. A limitation to the methods employed in this study is a lack of description of if and how model fit quality was evaluated. For the method of parametrizing neural power spectra: it is important to validate that models fit the data well, otherwise, the estimated parameters may be unreliable. This is especially important in developmental and clinical data, as analyzed here, as this data can be quite noisy, and differences in levels of noise across ages or between clinical groups could plausibly lead to differences in model fit quality. Useful quality checks for this kind of analysis would be to report the average r-squared (not just for the entire sample) for the parametrized data, and to examine whether model fit quality is significantly related to age, or clinical status and potential shows an overfit. Note that there is also a detailed guide for how best to apply spectral parametrization to developmental datasets, including notes on quality control, that may be useful: Ostlund, B., Donoghue, T., Anaya, B., Gunther, K. E., Karalunas, S. L., Voytek, B., & Pérez-Edgar, K. E. (2022). Spectral parameterization for studying neurodevelopment: How and why. *Developmental Cognitive Neuroscience*, 54, 101073. <https://doi.org/10.1016/j.dcn.2022.101073>.

Moreover, the modification of the parametrization: the authors should formally explain what has been done (rather than explaining which functions were adapted, which provides a bit of the impression that the authors did not fully understand what the changes in the code actually meant. Finally, what is the point of specifying peak_width_limits set to [0.5, 18.0], max_n_peaks = 24 7, and peak_threshold = 2, when the periodic power spectrum was determined by subtracting the FOOOF estimated aperiodic spectrum

Finally, the authors state correctly that the Schawaronkow paper only parametrized 1-10Hz due to extensive muscle artifacts. How can the authors ensure that their data set does not also suffer from typically observed infants muscle noise, which might be related to age changes as well and thus bias the current results? The statement "Our preprocessing artifact removal pipelines (see Methods) includes several steps for removing high-frequency noise from muscle artifact." is not sufficiently convincing. Additional sensitivity analyses could be conducted (e.g. conducting electrode-wise analyses, 1-10 Hz parametrization, providing model-fits, etc.).

Periodic activity (Power) calculation:

Using pre-defined canonical frequency bands instead of individualized frequency bands is highly problematic as it is known, that the alpha peak (and subsequently other frequency bands) change during development and show inter-individual differences. The same problem applies to the alpha coherence calculation. Also, manually selecting 30-second artifact-free segments for the alpha coherence seems to be a bit cherry-picking, why not selecting all artifact free segments and then randomly sample (e.g. permutation analysis).

"Whole brain power spectra for each individual were calculated by averaging across electrodes", while the supplementary demonstrates the results for 4 regions of interest (ROI) (frontal, central, temporal, and posterior). Why not conduct the analyses for each electrode and plot topoplots (and compute cluster-based permutation analyses).

In addition, the authors should carefully evaluate if the beta-band activity is not influenced by lower-frequency rhythms (i.e. harmonics), as presented in Schaworonkow et al., 2023.

Finally, Analysis for low beta peak association with anesthesia-induced frontal alpha coherence: The analyses for this research question are vastly unclear from the results section. It is unclear why only 23 infants (indicated in the figure) were analyzed from the 36 recorded infants (30% exclusion rate seems to be very high).

Statistical Analysis:

"Spectra were then averaged across individuals within 8 age bins." It should be clarified that this was done only for visualization (as I understood) but not for the actual statistical analyses

Moreover, it is unclear why ANCOVAs were conducted for the relationship between alpha coherence and the presence of low beta peak (and not a logistic regression, which additionally accounts for age and sex).

Interpretation:

The interpretation regarding the changes in periodic beta power are related to thalamocortical loop maturations purely speculative. No data for thalamocortical circuitry - purely speculative. There are other studies, which conducted DTI analysis to quantify the thalamocortical connectivity (Troendle et al. 2022).

"In addition, the aperiodic slope has been linked to the excitatory-inhibitory (E/I) balance of the underlying neuronal network, where a flattened, reduced slope is associated with increased excitation over inhibition, and a steeply more accelerated slope with increased inhibition over excitation." This is one possible explanation, but many alternative explanations are possible as well.

There are many oversimplified statements, for which there is insufficient evidence and potential alternative explanations (e.g. aperiodic offset might be influenced by many factors, such as physiological and non-physiological artifacts, skull thickness, etc.). Examples are:

"The aperiodic offset is hypothesized to represent broad band neuronal firing, and thus early increases in aperiodic offset are consistent with established increases in neuronal number, gray matter volume, and synaptic number during the first year."

"The majority of neural oscillations observed in the power spectrum are the direct result of inhibitory and thalamocortical network responses to sensory input."

"Similar to alpha oscillations, the generation of beta oscillations relies on GABAergic interneuron networks and thalamocortical connectivity." (reference is missing)

Minor comments:

Introducing new results in the Discussion (FXS infants) should be avoided and moved to the discussion section. The small sample size in the FXS group should be interpreted with caution or not at all. Finally, it is strange that the sample size is $N = 5$, but Figure 5B displays more than 5 lines.

Figure 1:

Figure 1A: This is rather an unnecessary figure. Why is offset defined at 2.5 Hz?

Figure 1C: removing the lines, would be better to illustrate the different time points, when each subject was assessed.

Figure 1G: it is difficult to dissociate cross-sectional from longitudinal effects here. Wrong color scale. Absolutely arbitrary zero (white color). For power, you must use unimodal color scales.
Figure H-I: These are the best subplots in these figures. I would recommend to additionally plot the longitudinal changes if individual subjects (and their average) as line plots (i.e. spaghetti plots) - and not just the modeled trajectories

"by 6-months only 15% of infants have two peaks in this range, and for most infants it is the higher 9.5Hz peak that is no longer observed. At 6-months fewer than 40%, of infants exhibit a dominant peak in the "alpha" (6.5-12Hz) range (Fig 1D) and the average peak frequency in the theta/alpha range is 6.3 ± 1 Hz. This disappearance of the higher peak after 4-months of age may reflect a transient step in thalamocortical circuitry development."

This is all purely descriptive and should be tested statistically to make a statement such as "disappearance". The same critique applies to the Transient beta peaks between 4-18 months.

I would recommend not to use "aperiodic power" as terminology. The better terminology is aperiodic activity.

Rephrase "choreographed sequence".

"1/f power law distribution (Fig 1A) and reflects non-oscillatory neuronal spiking activity". A reference is missing here.

I would recommend not presenting results already in the introduction (last section).

"Modeled developmental trajectories of the aperiodic offset showed a sharp linear increase over the first year after birth for both males and females (Fig 1H).

Modeled developmental trajectories of the aperiodic slope showed a gradual increase over the first year. These findings contrast with consistent reports of decreasing offset and slope across child and adulthood¹¹⁻¹⁴,..."

What defines a "sharp" and a "gradual" increase? Can the slopes be compared statistically?

Nature Communications NCOMMS-23-37310-T

Developmental trajectories of EEG aperiodic and periodic power: Implications for understanding the timing of thalamocortical development during infancy.

Dear ,

We thank you and the reviewers for a thoughtful review of our manuscript. We have made substantial revisions to the manuscript, including new analyses and figures. Reviewer comments are highlighted in gray with our responses below.

Reviewer #1

This manuscript by Wilkinson and colleagues is a good addition to the growing field of developmental studies examining the development of EEG aperiodic and periodic activity. They demonstrate non-linear changes in aperiodic and periodic power features in a large developmental dataset of resting-state EEG from 592 infants aged 2–44 months. The authors found changes in aperiodic offset and aperiodic slope, as well as periodic alpha, low beta, high beta, and gamma power. They link these aperiodic slope and alpha and beta power changes to the developmental progression of thalamocortical circuitry, framing these findings in the established literature and demonstrating the power of investigating these electrophysiological changes as a window to the underlying physiology. They also present evidence that the presence of low beta is associated with thalamocortical connectivity sufficient for anesthesia-induced alpha coherence.

While the manuscript is overall interesting, we have a number of comments and suggestions, listed below.

** Periodic changes reflect underlying developmental changes in thalamocortical circuitry: The major argument of the paper is that changes in aperiodic slope and changes in oscillatory alpha and beta oscillations reflect changes in the development of the underlying thalamocortical circuitry. There is thorough evaluation of the current literature and rationale for how this could be the case, and the preliminary evidence that low beta is associated with thalamocortical connectivity sufficient for anesthesia-induced alpha coherence helps make the argument. But given it is the primary argument of the paper, it would be nice to have another dataset that could be used to validate the larger hypothesis, or to soften the language on this topic.*

We thank the reviewer for their thorough review of our paper and their thoughtful comments and suggestions.

As much as we agree about the desirability of having another dataset to serve as confirmation for our findings, doing so is beyond the scope of the current paper. To address the reviewer's concerns, we have softened the language in several areas, as described below:

Introduction: "Consistent with our hypothesis, we find infants with an identifiable low beta peak have higher anesthesia-induced alpha coherence than those that do not, providing preliminary evidence that the emergence of this peak may be associated with thalamocortical loop maturation."

Discussion: "Second, more work is needed to firmly establish the association between changes in periodic power and development of thalamocortical circuitry. While we provide some preliminary evidence that the emergence of low beta is associated with maturation of thalamocortical connectivity, the dataset is small and still an indirect measure. Future research that combines EEG and MRI in infants from 4-6 months and again at 10-15 months, could provide more direct evidence linking alpha and beta periodic EEG findings with the development of thalamocortical connectivity."

** Within-subjects analysis: The high-powered, between-subject analysis demonstrates how these periodic changes in alpha and beta oscillations are non-linear and consistent with development of thalamocortical circuitry. Given that the data is longitudinal, the results would be strengthened if they could be also demonstrated within-subjects.*

While our dataset is longitudinal, our ability to examine developmental trajectories in an exclusively within-subjects fashion is limited by the fact that many participants do not have data from all timepoints. As can be seen in Figure 1C, any analysis restricted to participants with complete longitudinal data at any given pair of time-points would significantly reduce power. The approach we currently use (GAMMs with a random effect of participant) does leverage the longitudinal nature of the data, and is therefore not entirely between-subjects.

In addition, we now provide spaghetti plots of the full data set and of a subset of data limited to those participants with 4 or more EEG data points (Supplemental Figures 3, 4). These plots show similar within-subject trends to the GAMMs models.

** Page 3, lines 35-36: Sentence starting with "The majority of neural oscillations..." should have a citation. This is an important sentence for establishing how the aperiodic and periodic changes found reflect the underlying changes in thalamocortical circuitry, and should cite the first papers that established this link.*

Yes this was our oversight. Thank you for catching this. We have now added the following citations:

Buzsáki G, Chrobak JJ. Temporal structure in spatially organized neuronal ensembles: a role for interneuronal networks. *Curr Opin Neurobiol.* 1995 Aug;5(4):504-10. doi: 10.1016/0959-4388(95)80012-3. PMID: 7488853.

G. Buzsáki, L.W. Leung, C.H. Vanderwolf. Cellular bases of hippocampal EEG in the behaving rat. *Brain Research.*, 287 (1983)

Lopes da Silva F. Neural mechanisms underlying brain waves: from neural membranes to networks. *Electroencephalogr Clin Neurophysiol.* 1991 Aug;79(2):81-93. doi: 10.1016/0013-4694(91)90044-5. PMID: 1713832.

Hughes, S. W. & Crunelli, V. Thalamic mechanisms of EEG alpha rhythms and their pathological implications. *The Neuroscientist : a review journal bringing neurobiology, neurology and psychiatry* **11**, 357–372 (2005).

Hughes, S. W. & Crunelli, V. Just a phase they're going through: The complex interaction of intrinsic high-threshold bursting and gap junctions in the generation of thalamic α and θ rhythms. *International Journal of Psychophysiology* **64**, 3–17 (2007).

** Page 5, lines 7-8: While the necessary details are provided in the methods section, it would be nice to say how the spectra were calculated (i.e., multitapers).*

This now reads: "Whole brain power spectra for each individual were calculated using a multitaper spectral analysis for each electrode, and then averaging across electrodes."

** Figure 1A-B: It would be nice to see the full power spectrum first, then with the aperiodic and periodic components separated (as is the case in Figures 1D-F). Right now, it may make the reader think there is a way to separate the aperiodic and periodic signals through frequency decomposition instead of through spectral parameterization.*

We have edited Figure 1A to include both the absolute and aperiodic spectra. Thank you for this suggestion.

** Page 9, lines 4-5: The framing of a lack of a low beta peak as a "trough in low beta" seems unnecessary and confusing for two reasons. First, for there to be an apparent trough in low beta, there has to be peaks in alpha and high beta, but there can be a lack of a low beta peak regardless of alpha and high*

beta peaks. If oscillatory changes are meant to be independent (at least in principle), the terminology we use to describe them should not rely on how oscillations of different frequencies relate to each other. Second, a trough in low beta could also be caused by poor model fits where the aperiodic portion of the power spectrum is particularly noisy. For these reasons, I feel it would be better to avoid the trough terminology as a whole.

We also went back and forth on how to describe the lack of a low beta peak, especially as it relates to the poor fit of SpecParam. We have made edits to both the main text and methods”

Results: Transient beta peaks between 4-18 months:

“First, the shape of the periodic power spectra in the low beta range is notable for the absence of a low beta peak prior to 1 year of age (Fig 3A), with only 10% of infants (24/222) exhibiting a peak between 6-8 months of age (Fig 3A-C).”

Methods: EEG Power analysis):

“Specifically, the SpecParam modeled curves did not accurately capture the “trough” in the power spectra visually observed in this frequency range at younger ages (Supplemental Figure 4) when alpha and high beta peaks are prominent, but a low beta peak is not present.”

** Topographic plots of the different features would be nice to see, to get a sense of the rough spatial distributions.*

Thank you for this suggestion. Please see Supplemental Figure 6 for added topoplots.

** q-values are adjusted p-values, presumably FDR, but FDR details aren't given and this isn't really explicated.*

Additional detail has been added to the sentence in Statistical Analyses describing multiple comparisons correction, so it now reads:

“To correct for multiple comparisons, the false discovery rate (FDR) was controlled using the Benjamini and Hochberg method⁷⁶, which was applied for each model term within each region of interest across the 16 measure types (e.g., theta power, beta power, aperiodic offset) to produce q-values (FDR-corrected p-values).”

** It is unclear if coherence is across the two frontal bipolar pairs. Is this the case?*

** In addition, there are many ways of estimating coherence that focus on phase only, or just amplitude, or that include both. The authors need to include these details in the methods.*

Coherence was calculated between F7 and F8 (each electrode was referenced against their respective frontopolar electrode - Fp1 / Fp2). We now described in more detail the method used to calculate coherence between the 2 signals (Page 23, line 25):

“Coherence analysis was calculated between F7 and F8 using the multitaper method with the following parameters: window length T = 6s with no overlap, time- bandwidth product TW = 3, number of tapers K = 5, and spectral resolution of 2W = 1 Hz. We calculated coherence by quantifying the degree of correlation between both signals across a frequency range as previously described (Akeju Anesthesiology 2014):

$$C_{xy}(f) = \frac{|S_{xy}(f)|}{\sqrt{S_{xx}(f)S_{yy}(f)}}$$

Where $S_{xy}(f)$ is the cross-spectrum between the signals $x(t)$ and $y(t)$ (i.e., F7 and F8 electrodes), $S_{xx}(f)$ is the power spectrum of the signal $x(t)$ and $S_{yy}(f)$ is the power spectrum of the signal $y(t)$. Then, the median coherence within the alpha band was used for analyses.”

Please note that since the method is based on the cross-spectrum, and since we are estimating the imaginary component of the spectrum, phase is included in the Coherence calculation.

** There are significant non-neural changes in this age range as well, such as changes in skull thickness, the closure of the skull sutures, etc. that can affect the signals under investigation. These can't exactly be controlled for, but they need to be discussed as potential limitations.*

Yes, we agree. We have added the following to the discussion:

“..there are significant non-neural anatomical developmental changes that occur during infancy, including increases in skull thickness, gradual closure of fontanels and sutures, and changes in cerebrospinal fluid volume. Such changes can alter conductive properties of the skull, and in turn impact EEG signals.”

** Why is sex included as an ordered factor, instead of categorical? In terms of the output of the model I'm not sure it makes a difference in the math of the regressions, but I'm curious.*

The text has been updated to include the following: “oSex represents sex stored as an ordered factor; coding sex as an ordered factor is necessary for the GAMMs model to produce a single significance value for the age-by-sex interaction. This is more interpretable than the alternative of including sex as a categorical factor where two separate smooth effects of age are modeled for each sex, but no direct comparison is provided.”

Reviewer #2:

I appreciate the opportunity to review this manuscript. This study examined the longitudinal trajectories of periodic and aperiodic EEG power in a large, normative sample of infants. Results suggest non-linear increases in aperiodic offset and slope. Moreover, the manuscript examines the presence and changes in periodic alpha and beta, after accounting for the aperiodic slope. Changes in the presence (and absence) of alpha and beta peaks are interpreted as developmental changes in thalamocortical circuitry. This interpretation is further supported by evidence that the emergence of the low beta peak is associated with alpha coherence during anesthesia – an effect known to depend on thalamocortical connectivity. The study provides novel and important findings that add to our understanding on normative development of brain in infancy and toddlerhood by utilizing new ways to analyze EEG data. Moreover, the manuscript is well written and has several notable strengths such as utilizing a large group of infants followed longitudinally and a pharmacological manipulation. However, I have a couple questions and suggestions that I believe would improve the manuscript and its potential impact:

We thank the reviewer for their critical review of our manuscript and their support in the novelty of the findings presented.

1. The interpretation of beta power being related to infant movement is an important one. I do not agree with the authors that beta power would only be observed in frontal or occipital sites because movement-related beta is often seen in central sites (over motor cortex), as observed by the authors. I have two suggestions. One is that recent findings suggest that movement-related beta also changes with age in what I believe is a similar pattern as found by the authors (Rayson et al., 2023). If this is correct, this should alleviate concerns about the developmental effects being due to differences in movement, but even when matching for movements, similar developmental patterns in brain activity are observed. Second, a more labor-intensive solution would be to code some of the videos at different ages and show that the effects of interest are not explained by differences in movement.

Thank you for referring us to the Rayson et al., 2023. We were pleased to see similar findings in their periodic spectra at 9 and 12 months in the presence of the 30Hz beta peak, as well as emergence of the low beta peak at ~15Hz shown in their Figure 1. However, it seems that these power spectra were calculated from EEG data collected while the infants were moving (reaching, grasping, and shaking a toy), so we did not feel they were the ideal comparison as it relates to movement effects in our data.

As the reviewer suggested, we provided data from a subset of our sample and utilize behavioral scores given during baseline EEG collection of Study 4. In Supplemental Figure 8 we find similar

changes in low and high beta in infants with minimal movement (Behavior Score 4 and 5) and those with moderate movement (Behavior Score 2 and 3). In fact, those with least movement (Behavior Score 5) show the most consistency with Figure 1F). In addition, we did not find any significant correlation between Behavior Score and Low Beta Peak Amplitude (Pearson $R = 0.1$, $n=296$), or between Behavior Score and High Beta Peak Amplitude (Pearson $R = 0.027$, $n= 296$)

Supplemental Figure 8: Periodic power graphed by behavioral scores provided during EEG acquisition of Study 4. At the time of acquisition a behavioral score 1-5 was provided by the research assistant based on the infants behavioral regulation during baseline EEG acquisition, with higher scores indicating more behavioral compliance.

2. Combining multiple studies is a clear strength of the manuscript. However, it is not clear how the authors are accounting for the nesting of children within studies. I see that the authors are controlling for study as a fixed effect. However, is the co-dependence between observations between studies accounted for in any way in the GAMMs? I am not as familiar with GAMMs, but if it were an MLM, it could be accounted for by nesting the random effects at different levels (e.g., EEG assessments are nested within children, which are nested within studies). Please clarify if something like this was done by clearly outlining which fixed and random effects were estimated.

Originally, study was only considered as a fixed effect in this analysis. Both Reviewer 2 and 3 recommended changing this to random effects. We have updated our models, statistics and graphs to reflect these changes. Major findings were not altered.

3. I believe the authors are using previous versions of the HAPPE software to clean their data. The old version is known to overcorrect during artifact rejection, potentially taking away aspects of the signal. Why did the authors decide to use this method over the new version of the method (e.g., version 3; Monachino et al., 2022)? I believe the new algorithms fix many of the concerns from the previous version. It would be reassuring if the authors reprocessed at least some of the data and observed similar results or high correlations between the values used in the manuscript and ones reprocessed with newer versions of the software.

The original version of HAPPE was used for this analysis for several reasons. The original version of HAPPE embedded into BEAPP had been thoroughly tested in our lab using data from Study 2 (Gabard-Durnam et al. 2018). Using this pipeline, we have published several papers using longitudinal infant EEG data to successfully predict developmental outcomes. Given this, we felt confident that outputs from the pipeline provided meaningful neurobiological data related to development. In addition, we find that the EEG features described in this paper can be used to accurately predict an infant's age with a $R^2 = 0.82$ and mean average error of 92.4 days (Poster - An et. al. 2023; paper in prep) using a multilayer perceptron model. Together, we believe these findings further support the current pipeline's accuracy in estimating EEG features that represent each individual's underlying neurobiology.

We have also closely followed the development of HAPPE software including testing whether new versions can be used for our research aims. While the new version of HAPPE software has been thoroughly tested for ERP analyses - as described in Monachino et al., 2022, it has not yet been evaluated or optimized for spectral analysis. Importantly, in our testing of HAPPE V3, we have found that model fits were poor using SpecParam, even after utilizing our edited code, preventing a direct comparison of values used in the manuscript. Given, this we proceeded with using HAPPE 1.0. We do note that the Rayson 2023 paper the reviewer referenced earlier uses a different processing pipeline, and as the review notes, Figure 1 showed similar developmental

changes at 9 and 12 months in the beta band. We have added further discussion in a limitation section (Page 16)

“Third, the findings presented here utilize a specific artifact removal and processing pipeline (BEAPP and HAPPE1.0). An automated pipeline was used to improve reproducibility of our findings, and the code is available on Open Science Framework (<https://osf.io/u3gp4>). We do note that similar developmental changes in the low and high beta from 9 and 12 months are shown in Figure 1 of Rayson et al. 2023 which used a combination of MADE and NEAR pipelines. Interestingly, recently Rico-Pico et al. 2023 also described developmental changes in the periodic power spectra using MADE processing pipelines but limited their analysis to 1-20Hz “due to a power bump” in the gamma range, assumed to be related to muscle artifact, but likely the same high beta peak we observe most prominent at 7-8 months of age. Power spectra from 6, 9, and 16 months shown in Supplemental Figure 8 of Rico-Pico et al. 2023 visually show a similar pattern in beta band change across development. We hope that our comprehensive characterization of developmental changes using a single pipeline with frequent age sampling across the first 3 years of development spur other infant EEG researchers to evaluate the patterns we describe in their own data.”

Gabard-Durnam LJ, Mendez Leal AS, Wilkinson CL, Levin AR. The Harvard Automated Processing Pipeline for Electroencephalography (HAPPE): Standardized Processing Software for Developmental and High-Artifact Data. *Front Neurosci.* 2018 Feb 27;12:97. doi: 10.3389/fnins.2018.00097.

An WW, Bhowmik AC, Nelson CA, Wilkinson CL. Estimating chronological age from resting EEG power in children aged 0-3 years. *Society for Neuroscience*, 2023.

4. Several statements are missing a citation. For example, “The majority of neural oscillations observed in the power spectrum are the direct result of inhibitory and thalamocortical network responses to sensory input.” Or “The thalamus is thought to play a central role in the generation of the mature posterior alpha rhythm.”

We apologize for this oversight. Citations have been added.

5. For Figure 1, it is unclear how aspects of the figure are related to the analyses or if they represent real data. For example, the authors display the data in age bins, but as far as I understand, the analyses were conducted using continuous age. It is unclear if Panels A and B display simulated data as an example, one participant, one age group, or the average of all participants – please clarify these points in the figure legend.

We have edited the Figure legend as follows:

Figure 1. Developmental Trajectories of Aperiodic and Periodic Power Spectra

(A, B) Example power spectra derived from 6-8 month-old participants from Study 2. Mean spectra with shaded errors describing 95% confidence intervals. **(A)** Aperiodic offset is defined as power at 2.5Hz. **(B)** Periodic power peaks defined as maxima within a defined frequency range. Periodic band power defined as the integral of the periodic power spectra between defined frequencies. **(C)** Longitudinal study enrollment. Each line is a participant with dots indicating when EEG was collected for that participant. **(D-F)** Absolute, Aperiodic, and Periodic power visualized across 8 age bins (see Table 1). Spectra from EEGs collected at within each age bin were averaged and shading describes 95% confidence intervals. **(G)** Heatmap showing age-related changes in periodic power binned every 2 months. **(H-I)** GAMMs modeled trajectories of aperiodic offset and slope for males (orange) and females (blue). Here, age is incorporated into the model using exact age in days, rather than age-bins. Relative inflection points are shown with circular markers. Below, heatmaps show the standardized change in offset or slope per day, defined as [change per day]/[standard deviation of measures across full age range]

6. Is it possible that the early alpha peak transitions to the low beta peak? I was not able to tell as 4-6 months was not displayed on Figure 2.

We do not believe this to be the case as the proportion of infants with an early alpha peak between 6.5-12Hz dramatically falls after 2-4 months of age (Figure 2D), and from 4-8 months the vast majority of infants do not yet have a low beta peak (Figure 3C), which begins to emerge sometime after 8 months of age. This “gap” suggests that these are distinct peaks rather than a transition from early alpha to low beta. This can also be seen in the individual plots shown in Supplemental Figure 2, and the average power spectra plots by age bin in Figure 1F.

7. For the analyses with anesthesia, are the authors analyzing a difference in coherence between before and during anesthesia? Namely, is the outcome of the ANCOVA a change score? Or how is the baseline recording being used?

No. To clarify, we are analyzing whether anesthesia-induced alpha coherence is higher in infants with or without a low beta peak detected in the baseline awake state EEG (prior to any anesthesia administration). We do not assess pre-anesthesia alpha coherence.

8. I appreciate the supplemental analyses by ROI, but it would be useful to provide topoplots for the

effects of interest. Moreover, some of the labels for the figures in the supplement are missing (Figure S3A, S3B)

Please see added Supplemental Figure 4 for topoplots added for visualization ROI differences. We have also added labels for S3A and S3B.

9. What do the dotted lines stand for in Figure 4B?

These represent quartiles. We have added this to the figure legend.

10. It is unclear how many infants had longitudinal data, and how many assessments each infant had. Please provide more details on exactly how many participants had longitudinal data and for which ages. A suggestion would be to create an additional table.

We have added to Table 1 the % of infants with longitudinal data at each age bin and provided the average number of EEGs from each participant.

11. For Table 2, please add number/percent of segments retained after preprocessing.

Apologies for the confusion. The first row of Table 2 already shows the number of segments retained after processing. We have clarified the table heading.

12. Minor: For Table 1, it is strange to say that infants have income. I would suggest saying families rather than infants.

Corrected.

13. Minor: The name of FOOOF has been updated to SpecParam. Please use the updated name.

Corrected.

14. Minor: Please include page numbers.

Corrected.

References:

Monachino, A. D., Lopez, K. L., Pierce, L. J., & Gabard-Durnam, L. J. (2022). The HAPPE plus Event-Related (HAPPE+ER) software: A standardized preprocessing pipeline for event-related potential analyses. *Developmental Cognitive Neuroscience*, 57, 101140. <https://doi.org/10.1016/j.dcn.2022.101140>

Rayson, H., Szul, M. J., El-Khoueiry, P., Debnath, R., Gautier-Martins, M., Ferrari, P. F., Fox, N., & Bonaiuto, J. J. (2023). Bursting with potential: How sensorimotor beta bursts develop from infancy to adulthood (p. 2023.05.09.539976). *bioRxiv*. <https://doi.org/10.1101/2023.05.09.539976>

Reviewer #3 :

The study titled "Early Development of Neural Circuits in Infants Revealed Through EEG Analysis" presents an investigation into the development of aperiodic and periodic brain activity in infants. The strength of the current study is its large sample size. However, there seems to be a lack of substantial novelty for such a high-impact paper as Nature Communications. Studies on longitudinal changes of aperiodic and periodic activity have been previously published (e.g. Schaworonkow & Voytek, 2021, Rico-Pico et al., 2023). The actual novelty of the paper is the investigation of the relationship between low beta peak developmental emergence of anesthesia-induced frontal alpha coherence (within a small sample).

We appreciate the reviewer's critical and thorough review of our paper. As the reviewer rightly points out previous studies have investigated longitudinal change in aperiodic and periodic power in infancy, however we believe that both the sample size and sampling frequency across the 2-44 month age range of our analysis enabled the observation and analysis of novel findings not previously described in the literature. As we discuss in the paper, Schaworonokow & Voytek 2021 was limited to 21 infants aged 30-210 days and spectral analysis was limited to the 1-10Hz range - thus changes in the beta band were not analyzed. Similarly, Rico-Pico et al., also limited their spectral analysis to 1-20Hz frequency range "due to a power bump in the gamma band range" which led to poor SpecParam model fit. Indeed this power "bump" is likely the same as what we describe as a high beta peak. Rico-Pico et al. was limited by data between 6 and 18 months, so they did not have the ability to observe the subsequent reduction in the high beta peak amplitude and frequency, presumably leading them to assume the "bump" was due to muscle artifact rather than developmental changes in brain activity. We now also include this reference in our paper.

More importantly, there are several aspects (analytical procedures) of the study that warrant critical examination (see more details below). As infant data usually exhibit lower signal quality (due to excessive motion artifacts and limited cooperation), the data and preprocessing need to be evaluated carefully and additional control analyses and visualization should be provided to ensure the validity of the results and its interpretation.

We very much agree that infant data requires careful analyses and visualization and appreciate the reviewer's critical examination. Please see our detailed answers below.

EEG acquisition:

The site should be included as random effect in the analysis. Even if the same equipment was used (for 3 out of 4 studies), it is expected that large site effects are observable (e.g. due to distinct technical personnel).

Thank you for this recommendation. We have updated our models with study now included as a random effect, statistics and graphs to reflect these changes. Major findings were not altered.

Impedances were kept below 100kΩ is even very high for a Geodesic NetAmps amplifier.

We appreciate the reviewer's concern here. We have previously spoken to EGI representatives at the beginning of the studies, and based on our high-impedance amplifier in an electrically-shielded testing room, we decided to use the 100kΩ threshold given the limited time frame of adjusting impedances before infants and toddlers become fussy.

EEG preprocessing:

The MARA algorithm is not trained for infants data and thus inadequate choice in the current analysis pipeline. From personal experience, I know that the ICA MARA algorithm can remove insufficient amount of components or possible also being too conservative (removing too many components), which might be the case here: on average ~35 components were removed, which is approximately half-of the components included in the ICA. Consider using ICLabel or do manual inspection of the components. The fact that Moreover, it is unclear how the additional electrodes (MARA: 64-channel net – 16, 9, 8, 3, 58, 57, 21, 25, 18, 22 30, 43, 50, 53, 32, 33, 38, 41, 45; and 128-channel net - 28, 19, 4, 117, 13, 112, 41, 47, 37, 55, 87, 103,23 98, 65, 67, 77, 90, 75.) were implemented in MARA.

To clarify, on average 35% of components were removed, not half. Table 2 states “Percent ICs rejected” not Number ICs rejected. While the reviewer is correct that the MARA was not trained on infant data, the use of MARA in context of the HAPPE pipeline, which first implements a wavelet threshold step, was thoroughly tested on infant data - described in Gabard-Durnam et al 2018. We wonder whether the reviewer has utilized ICA MARA with the prior wavelet threshold step built into HAPPE? Gabard-Durnam et al., tested artifact rejection performance between the full HAPPE pipeline utilizing wavelet thresholding followed by ICA with MARA vs. ICA with MARA alone. HAPPE (wavelet thresholding preceding ICA with MARA) increased the % EEG signal retained from ~25% to ~70%.

Here is the information provided from Gabard-Durnam et al.: “MARA was not trained specifically on developmental or patient data, but several findings support its application in these contexts. First, the anatomical correlations of the 10–20 electrodes that MARA uses to calculate its spatial features are highly consistent across infant and adult brains (Kabdebon et al., 2014). That is, comparable information is supplied to MARA's spatial features from the 10–20 electrodes regardless of age. Second, a potential concern during HAPPE development was

that one of MARA's spectral features evaluates EEG alpha band power, where very young infants or clinical populations may show a different frequency power peak than the alpha band observed in healthy adults (Stroganova et al., 1999; Lansbergen et al., 2011). However, empirically, even the youngest infants tested (3 months of age) in the present dataset had enough alpha power in the components to make use of this MARA criterion appropriately. Indeed, the 3-month files in the example analysis below had the lowest rates of MARA component rejection in the sample. Variation in the alpha band peaks was also preserved across the developmental datasets and was consistent within individual files before and after MARA component rejection, suggesting alpha peaks were unperturbed during the component rejection algorithm. Lastly, the rates of MARA component rejection for the datasets run through HAPPE were comparable to both the rejection rates for the adult data used to validate MARA (Winkler et al., 2011, 2014) and to rates of manual component rejection with developmental data decomposed with ICA (Piazza et al., 2016). HAPPE therefore includes MARA as a robust evaluation tool for component rejection suitable for developmental and clinical data.”

MARA is integrated into the HAPPE code allowing for incorporation of additional electrodes and thus analysis of additional components to determine probability of a component being artifact and automatically removed (based on determined threshold). Users are guided to limit the number of electrodes added based on the length of EEG recording and sampling rate to meet requirements for reliable ICA decomposition.

Reference: Gabard-Durnam LJ, Mendez Leal AS, Wilkinson CL, Levin AR. The Harvard Automated Processing Pipeline for Electroencephalography (HAPPE): Standardized Processing Software for Developmental and High-Artifact Data. *Front Neurosci.* 2018 Feb 27;12:97. doi: 10.3389/fnins.2018.00097.

FOOOF parametrization:

The description of the FOOOF parametrization is insufficient. A limitation to the methods employed in this study is a lack of description of if and how model fit quality was evaluated. For the method of parametrizing neural power spectra: it is important to validate that models fit the data well, otherwise, the estimated parameters may be unreliable. This is especially important in developmental and clinical data, as analyzed here, as this data can be quite noisy, and differences in levels of noise across ages or between clinical groups could plausibly lead to differences in model fit quality. Useful quality checks for this kind of analysis would be to report the average r-squared (not just for the entire sample) for the parametrized data, and to examine whether model fit quality is significantly related to age, or clinical status and potential shows an overfit. Note that there is also a detailed guide for how best to apply spectral parametrization to developmental datasets, including notes on quality control, that may be useful: Ostlund, B., Donoghue, T., Anaya, B., Gunther, K. E., Karalunas, S. L., Voytek, B., & Pérez-Edgar, K.

E. (2022). Spectral parameterization for studying neurodevelopment: How and why. Developmental Cognitive Neuroscience, 54, 101073. <https://doi.org/10.1016/j.dcn.2022.101073>.

We completely agree with the reviewer in validating model fits, especially with infant data. This is in fact what led us to making edits to SpecParam. In the originally submitted paper we provide both the Mean R^2 for the full sample, as well as the full range across the sample. All data included had an R^2 above 0.89. In addition, in our supplemental data we provide the graphs of the average SpecParam model fit for each age bin, as well as the error across frequencies for 5, 6, and 7 months, where the highest degree of error was observed. We have made this clearer in the methods section:

“Code is available (osf.io/u3gp4) which runs both the original and modified versions of SpecParam, and graphs the RMSE across frequencies for both versions, separate by age. Comparisons model fits for each age for are presented in Supplemental Figure 4, as well as RMSE across frequencies for infants 5, 6, or 7 months old. Further analyses were subsequently restricted to 2.5-50Hz given elevated error between 2-2.5, and 50-55Hz. Mean R^2 for the full sample using this modified version of SpecParam was 0.997 (STD 0.008; range 0.890-0.999). Mean R^2 for each age bin ranged from 0.991-0.999. Mean estimated error for the sample was 0.01 (STD 0.01, range 0.002-0.09).”

Moreover, the modification of the parametrization: the authors should formally explain what has been done (rather than explaining which functions were adapted, which provides a bit of the impression that the authors did not fully understand what the changes in the code actually meant.

Thank you for raising this concern. We have edited the methods to provide more explanation of modifications made:

“Specifically, the SpecParam modeled curves did not accurately capture the “trough” in the power spectra visually observed in this frequency range at younger ages (Supplemental Figure 4) when alpha and high beta peaks are prominent, but a low beta peak is not present. In the original SpecParam, the power spectrum is first fit with an estimated aperiodic component which is subtracted from the raw signal. Any negative data, signal falling below 0, is converted to 0, and peaks are identified through iterative gaussian fits. Once all peaks are identified, these peaks are removed from the raw spectra and a final aperiodic component is fit. Between 2-7 months, signal between 10-20Hz often falls below the first estimated aperiodic fit and is thus converted to 0, impacting both the periodic peaks identified and the final fit of the aperiodic component. To improve model fit, the `robust_ap_fit` function, which initially defines the aperiodic component, was modified so that the initial estimate of the flattened power spectra

(flatspec) has a baseline elevated such that the lowest point is ≥ 0 , to avoid omitting data important across the 2-7 month age range.”

Finally, what is the point of specifying peak_width_limits set to [0.5, 18.0], max_n_peaks = 7, and peak_threshold = 2, when the periodic power spectrum was determined by subtracting the FOOOF estimated aperiodic spectrum

The final aperiodic component is estimated after peaks have been identified and thus parameters around peak fitting are relevant to the final aperiodic fit. While we do not use the estimated periodic spectrum from FOOOF/SpecParam, we felt it was important to provide correct peak parameters for the data being analyzed. This is also important for replicability of the findings by others using their own datasets.

Finally, the authors state correctly that the Schawaronkow paper only parametrized 1-10Hz due to extensive muscle artifacts. How can the authors ensure that their data set does not also suffer from typically observed infants muscle noise, which might be related to age changes as well and thus bias the current results? The statement "Our preprocessing artifact removal pipelines (see Methods) includes several steps for removing high-frequency noise from muscle artifact." is not sufficiently convincing. Additional sensitivity analyses could be conducted (e.g. conducting electrode-wise analyses, 1-10 Hz parametrization, providing model-fits, etc.).

We now provide additional sensitivity analysis from a subset of our sample, utilizing behavioral scores given during baseline EEG collection of Study 4. In Supplemental Figure 8 we find similar changes in low and high beta in infants who scored high (minimal movement) and those that scored lower (moderate movement). We note that infants with least movement (Behavior Score 5) show the most consistency with Figure 1F. In addition, we did not find any significant correlation between Behavior Score and Low Beta Peak Amplitude (Pearson $R = 0.1$, $n=296$), or between Behavior Score and High Beta Peak Amplitude (Pearson $R = 0.027$, $n= 296$)

Supplemental Figure 8: Periodic power graphed by behavioral scores provided during EEG acquisition of Study 4. At the time of acquisition a behavioral score 1-5 was provided by the research assistant based on the infants behavioral regulation during baseline EEG acquisition, with higher scores indicating more behavioral compliance.

Periodic activity (Power) calculation:

Using pre-defined canonical frequency bands instead of individualized frequency bands is highly problematic as it is known, that the alpha peak (and subsequently other frequency bands) change during development and show inter-individual differences. The same problem applies to the alpha coherence calculation. Also, manually selecting 30-second artifact-free segments for the alpha coherence seems to be a bit cherry-picking, why not selecting all artifact free segments and then randomly sample (e.g. permutation analysis).

Alpha coherence under anesthesia does not have the same developmental change in peak frequency as does alpha coherence during awake periods. This is shown in Cornelissen et al 2018, where anesthesia-induced alpha coherence begins to emerge at 10 months and is stable between 8-12Hz from 10 to 40 months of age (See Fig 4 in referenced paper).

To clarify, all available 30 second segments that were both under stable sevoflurane concentration and were artifact free were used in the analysis. We did not randomly sample as the measure of alpha coherence under anesthesia is generally stable within subjects (mean 0.33, SD 0.06 for our sample).

Cornelissen L, Kim SE, Lee JM, Brown EN, Purdon PL, Berde CB. Electroencephalographic markers of brain development during sevoflurane anaesthesia in children up to 3 years old. *Br J Anaesth*. 2018 Jun;120(6):1274-1286. doi: 10.1016/j.bja.2018.01.037. Epub 2018 Apr 5. PMID: 29793594; PMCID: PMC6617966.

"Whole brain power spectra for each individual were calculated by averaging across electrodes", while the supplementary demonstrates the results for 4 regions of interest (ROI) (frontal, central, temporal, and posterior). Why not conduct the analyses for each electrode and plot topoplots (and compute cluster-based permutation analyses).

EEG net placement on infants is not as exact or consistent across participants compared to cooperative adults (or older children). Given this, we chose to average across electrode regions. We also limited the electrodes chosen for analysis to those on both a 64 and 128 channel net, with the goal of our findings being applicable to those collecting data on EEG systems with fewer electrodes.

We have provided topoplots using averages from 6 regions of interest (left frontal, right frontal, central, left temporal, right temporal, and posterior) now in Supplemental Figure 6.

In addition, the authors should carefully evaluate if the beta-band activity is not influenced by lower-frequency rhythms (i.e. harmonics), as presented in Schaworonkoff et al., 2023.

We did consider this, and were reassured by the striking differences in longitudinal trajectories of alpha frequency (steadily increasing with age) compared high beta (non-linear with increases and decreases with age). Specific to the low beta peak, this peak emerges well after the alpha peak is present, such that the majority of infants between 11-15 do not have a low beta peak, despite having an alpha peak - making harmonics an unlikely cause of the low beta peak. When we restrict our analysis to 11-15 months infants with the highest quartile of alpha peak amplitude (n=58), 34% (n=38) have no low beta peak identified.

Finally, Analysis for low beta peak association with anesthesia-induced frontal alpha coherence: The analyses for this research question are vastly unclear from the results section. It is unclear why only 23 infants (indicated in the figure) were analyzed from the 36 recorded infants (30% exclusion rate seems to

be very high).

We have provided additional clarification for this analysis.

“Here we hypothesized that the presence of low beta oscillations (as measured by the presence of a low beta peak) during the infant’s awake, unanesthetized state would be associated with increased GABA-dependent anesthesia-induced frontal alpha coherence. EEG data from 36 infants (6-15 months old) were collected during the awake and anesthetized states. We first analyzed the baseline awake state EEG to confirm that similar developmental changes were observed. Developmental changes in the periodic power spectra in this smaller dataset were qualitatively similar to those described above (Figure 4A), with a low beta peak beginning to emerge in some infants after 7 months and present in roughly half the infants between 7-12 months of age (11/21). To test our above hypothesis, we then limited our coherence analysis to this 7-12 month age range, comparing alpha coherence in those with and without a low beta peak. As hypothesized, alpha coherence was significantly increased in those with a low beta peak compared to those without (ANCOVA, with sevoflurane level as covariate; $F(1,20) = 5.25$, $p = 0.03$; Figure 4B).

Statistical Analysis:

"Spectra were then averaged across individuals within 8 age bins." It should be clarified that this was done only for visualization (as I understood) but not for the actual statistical analyses

We have edited this as follows: “ For visualization of developmental changes, spectra were then averaged across individuals within 8 age bins.”

Moreover, it is unclear why ANCOVAs were conducted for the relationship between alpha coherence and the presence of low beta peak (and not a logistic regression, which additionally accounts for age and sex).

Here we hypothesize that the presence of a low beta peak is predictive of anesthesia-induced alpha coherence as a continuous variable, which is best assessed via ANCOVA. A logistic regression would assess a similar, but converse question - whether anesthesia induced alpha coherence was predictive of the presence of a beta peak. Given that we need account for the effects of sevoflurane levels on alpha coherence, and our hypothesis was focused on predicting alpha coherence, an ANCOVA made the most sense. Given our small sample size, we were not well powered to also include covariates age and sex.

Interpretation:

The interpretation regarding the changes in periodic beta power are related to thalamocortical loop maturations purely speculative. No data for thalamocortical circuitry - purely speculative. There are other studies, which conducted DTI analysis to quantify the thalamocortical connectivity (Troendle et al. 2022).

We have added this as a limitation to our discussion: “Second, more work is needed to firmly establish the association between changes in periodic power and development of thalamocortical circuitry. While we provide some preliminary evidence that the emergence of low beta is associated with maturation of thalamocortical connectivity, the dataset is small and still an indirect measure. Future research that combines EEG and MRI in infants from 4-6 months and again at 10-15 months, could provide more direct evidence linking alpha and beta periodic EEG findings with the development of thalamocortical connectivity.”

"In addition, the aperiodic slope has been linked to the excitatory-inhibitory (E/I) balance of the underlying neuronal network, where a flattened, reduced slope is associated with increased excitation over inhibition, and a steeply more accelerated slope with increased inhibition over excitation." This is one possible explanation, but many alternative explanations are possible as well.

There are many oversimplified statements, for which there is insufficient evidence and potential alternative explanations (e.g. aperiodic offset might be influenced by many factors, such as physiological and non-physiological artifacts, skull thickness, etc.). Examples are:

"The aperiodic offset is hypothesized to represent broad band neuronal firing, and thus early increases in aperiodic offset are consistent with established increases in neuronal number, gray matter volume, and synaptic number during the first year."

Additional references added. We have added the following section on limitations:

“First, there are significant non-neural anatomical developmental changes that occur during infancy, including increases in skull thickness, gradual closure of fontanels and sutures, and changes in cerebrospinal fluid volume. Such changes can alter conductive properties of the skull, and in turn impact EEG signals, including measures such as aperiodic offset and slope. Second, more work is needed to firmly establish the association between changes in periodic power and development of thalamocortical circuitry. While we provide some preliminary evidence that the emergence of low beta is associated with maturation of thalamocortical connectivity, the dataset is small and still an indirect measure. Future research that combines EEG and MRI in infants from 4-6 months and again at 10-15 months, could provide more direct evidence linking alpha and beta periodic EEG findings with the development of thalamocortical connectivity.”

"The majority of neural oscillations observed in the power spectrum are the direct result of inhibitory and thalamocortical network responses to sensory input."

References added.

"Similar to alpha oscillations, the generation of beta oscillations relies on GABAergic interneuron networks and thalamocortical connectivity." (reference is missing)

References added.

Minor comments:

Introducing new results in the Discussion (FXS infants) should be avoided and moved to the discussion section. The small sample size in the FXS group should be interpreted with caution or not at all. Finally, it is strange that the sample size is $N = 5$, but Figure 5B displays more than 5 lines.

We recognize that it is unconventional to include results in a discussion section, and are willing to shift it to the results if the editor and other reviewers feel it is important. For reader clarity, we felt that it was best included in the discussion as it relates to our interpretation of the transient high beta peaks observed in the longitudinal analysis. We felt that including the FXS finding in the discussion was a better flow than moving the discussion/interpretation of the beta peaks to the results section.

Regarding sample size, Figure A is restricted to children between 35-48 months old, whereas Figure B has data from individuals in a broader age range from 32-84 months old. We have edited the Figure legend to make this clearer.

Figure 1:

Figure 1A: This is rather an unnecessary figure. Why is offset defined at 2.5 Hz?

We included this figure, in part to make sure readers were aware that our offset was defined at 2.5hz. The reasoning is included in our methods section: "As the FOOF-determined offset is extrapolated to the estimated aperiodic power at 0Hz, where there are high amounts of error, we instead calculated the aperiodic offset based on aperiodic power at 2.5hz (Figure 1B)."

Figure 1C: removing the lines, would be better to illustrate the different time points, when each subject was assessed.

We found that by removing the lines it was much more difficult to see repeated visits on the individual level. Instead we have increased the size of the dots. Note that each line represents an individual participant so provide additional information about the longitudinal aspect of the study.

Figure 1G: it is difficult to dissociate cross-sectional from longitudinal effects here. Wrong color scale. Absolutely arbitrary zero (white color). For power, you must use unimodal color scales.

Figure H-I: These are the best subplots in these figures. I would recommend to additionally plot the longitudinal changes if individual subjects (and their average) as line plots (i.e. spaghetti plots) - and not just the modeled trajectories

Thank you for this suggestion. We have changed the figure to a perceptually uniform sequential colormap. Spaghetti plots are now shown in Supplemental Figures. We provide spaghetti plots of all data, as spaghetti plots limited to participants with at least 4 longitudinal data points.

"by 6-months only 15% of infants have two peaks in this range, and for most infants it is the higher 9.5Hz peak that is no longer observed. At 6-months fewer than 40%, of infants exhibit a dominant peak in the "alpha" (6.5-12Hz) range (Fig 1D) and the average peak frequency in the theta/alpha range is $6.3 \pm 1\text{Hz}$. This disappearance of the higher peak after 4-months of age may reflect a transient step in thalamocortical circuitry development."

This is all purely descriptive and should be tested statistically to make a statement such as "disappearance". The same critique applies to the Transient beta peaks between 4-18 months.

We have added additional analyses using generalized linear mixed effect models to evaluate these claims. Additional Results text reads:

"Age was significantly negatively associated with the probability of having two peaks across the 2-6 month age range using mixed effects logistic regression ($p < 0.0001$)."

"Mixed effects logistic regression demonstrated that the likelihood of having a low beta peak increased with age across the full age range ($p < 0.0001$)."

Additional Methods text reads:

Developmental changes in the presence or absence of peaks within specific frequency bands (i.e., broad alpha and low beta) were assessed using generalized linear mixed effects models using a binomial family with a logit link function using the lme4 package in R (Bates et al., 2014), of the form:

(1) $\text{Peaks} \sim \text{age_days} + (1 | \text{Study/New_ID})$, family = binomial(link = "logit")

In the model describing broad alpha peaks, data from the 2-6 month bins were included, and Peaks is a binary variable indicating whether the participant has two peaks within the broad alpha range or not. In the model describing low beta peaks, data from the full age range was included, and Peaks is a binary variable indicating whether a low beta peak was present or not. The models were specified with a random intercept, with participants nested within studies.

I would recommend not to use "aperiodic power" as terminology. The better terminology is aperiodic activity.

We have edited this in the results section.

Rephrase "choreographed sequence".

We have removed "choreographed" from this sentence.

"1/f power law distribution (Fig 1A) and reflects non-oscillatory neuronal spiking activity". A reference is missing here.

The following citations have been added. Thank you for catching this.

Manning JR, Jacobs J, Fried I, Kahana MJ. Broadband shifts in local field potential power spectra are correlated with single-neuron spiking in humans. *J Neurosci*. 2009 Oct 28;29(43):13613-20. doi: 10.1523/JNEUROSCI.2041-09.2009. PMID: 19864573; PMCID: PMC3001247.

Miller, K. J., Sorensen, L. B., Ojemann, J. G., & den Nijs, M. (2009). Power-Law scaling in the brain surface electric potential. *PLoS Computational Biology*, 5(12), e1000609.

Gao, R., Peterson, E. J., & Voytek, B. (2017). Inferring synaptic excitation/inhibition balance from field potentials. *NeuroImage*, 158, 70–78.

I would recommend not presenting results already in the introduction (last section).

This is a stylistic preference, and commonly done in Nature Communications and Nature Neuroscience. Unless the editor recommends otherwise, we prefer not to make changes to this section of the introduction.

"Modeled developmental trajectories of the aperiodic offset showed a sharp linear increase over the first year after birth for both males and females (Fig 1H).

Modeled developmental trajectories of the aperiodic slope showed a gradual increase over the first year. These findings contrast with consistent reports of decreasing offset and slope across child and adulthood^{11–14},...

What defines a "sharp" and a "gradual" increase? Can the slopes be compared statistically?

These are qualitative descriptions that are also further visualized in the standardized changes per day. For example for aperiodic offset and slope, standardized change in each measure per day is greater for offset than slope over the first year. We did not feel that slope and offset needed to be statistically compared our research aims were not seeking to determine whether there were differences in trajectories between these measures. We have added the word “qualitatively” to the text to make this more clear.

Reviewer #1 (Remarks to the Author):

The authors have done a very thorough job at responding to our comments and recommendations. We appreciate the softened language regarding the maturation of thalamocortical development, and find this to be a sufficient change. Also, the two comments we made regarding the statistical models — ordered effect of sex and within-subjects analyses — totally make sense. Those were silly oversights on our parts regarding the mixed effect models used.

Reviewer #1 (Remarks on code availability):

Because everything is organized into numbered computational notebooks, it was easy to follow without an explicit README. Everything looks straightforward.

Reviewer #2 (Remarks to the Author):

I appreciate the authors for being responsive to the reviewers' feedback. I think the manuscript has been greatly improved. I only have a couple minor comments.

1. For the supplemental analyses involving motion, it would be important to add details on how many infants and EEG sessions were coded – was it all of the data for only one study? It would also be important to provide more details on the coding scheme that was followed by the research assistants as well as any details on the training and reliability of the coding scheme.
2. For the anesthesia analyses, I still think it is confusing to mention on p. 22 a baseline period (pre-anesthesia) that was never analyzed in the current manuscript, but then mention that "baseline EEG" were preprocessed and analyzed for that analysis. For example, "Baseline EEGs (n = 45) were visually inspected and..." Please clarify in the manuscript that the baseline period was not analyzed and only the period during anesthesia was utilized in the analyzes presented in manuscript.
3. The authors should also cite the recent manuscripts (Rayson et al., 2023; Rico-Pico et al., 2023) that find a similar pattern of results, as they do in the letter.
4. Similarly, in the letter, the authors mention adding a limitation on p. 16, which is not in the revised manuscript. Specifically, the following paragraph (mentioned in the letter) is not present in the manuscript: "Third, the findings presented here utilize a specific artifact removal and..." Please add that entire section to the manuscript.

Reviewer #3 (Remarks to the Author):

I would like to thank the authors for addressing my and the other reviewers comments. Some of the (others and mine) comments were answered correctly and provided good explanation. However, other responses do not provide sufficient justification or evidence to the initial comments. In the following address them specifically:

I thoroughly agree with the reviewer #1 in terms of "But given it is the primary argument of the paper, it would be nice to have another dataset that could be used to validate the larger hypothesis, or to soften the language on this topic."

The authors softened the language, but the statements regarding the thalamocortical connectivity are still purely speculative. The data and results this study simply do not provide this information. Thus such claims in the title and abstract are not valid.

Abstract:

"We present preliminary evidence that the emergence of the low beta peak is associated with higher thalamocortical-dependent, anesthesia-induced alpha coherence"

Introduction:

"Consistent with our hypothesis, we find infants with an identifiable low beta peak have higher anesthesia-induced alpha coherence than those that do not, providing preliminary evidence that the emergence of this peak may be associated with thalamocortical loop maturation."

Discussion:

"While we provide some preliminary evidence that the emergence of low beta is associated with maturation of thalamocortical connectivity, the dataset is small and still an indirect measure."

It's not even about the small sample size, which is a major problem as I stated in my initial reviews, but rather that there might be several competing alternative explanations of the emergence of low beta rather than maturation of thalamocortical connectivity. Such a claim is only warranted if additional data (e.g. DTI) would demonstrate such a relationship (e.g. as in Troendle et al. 2022: <https://elifesciences.org/articles/77571>).

Moreover, the response to reviewer #1 valid suggestion to analyze the data in a within-subject design surprised me quite a bit: "While our dataset is longitudinal, our ability to examine developmental trajectories in an exclusively within-subjects fashion is limited by the fact that many participants do not have data from all timepoints." From Figure 1C, I speculate that the majority of the sample has at least 4 time points, which would qualify such a within-subject analysis.

To the comment of reviewer #2: "I believe the authors are using previous versions of the HAPPE software to clean their data. The old version is known to overcorrect during artifact rejection, potentially taking away aspects of the signal. Why did the authors decide to use this method over the new version of the method." The authors responded that "Importantly, in our testing of HAPPE V3, we have found that model fits were poor using SpecParam, even after utilizing our edited code, preventing a direct comparison of values used in the manuscript. Given, this we proceeded with using HAPPE 1.0." How such a discrepancy between the two pipelines in terms of SpecParam fit is explainable? As the reviewer #2 correctly states is that the HAPPE software overcorrects the data (which I also stated in my initial review with regard to the % of bad ICA by MARA).

Regarding the responses to my comments:

I have used the Geodesic NetAmps for >10 years and I know using an impedance threshold below 100kOhm would not provide us good quality of data even in healthy children and adults, for which movement artifacts are less of a problem as in infants. While I understand that adjusting impedances makes the infants become fuzzy, the signal of interest is very sensitive to good signal quality, which in my opinion is not given in this data set.

The reasoning about the preprocessing (ICA MARA, HAPPE pipeline) are further not convincing, mainly because of the validation approach in the Gabard-Durnam et al 2018 paper is rather inaccurate, as they do not have a true ground truth what the EEG signal is and thus can not really claim that % EEG signal retained from 25-70%.

Moreover, some of the authors on this paper are also authors of the HAPPE pipeline and thus an independent evaluation of the MARA (preceding wavelet thresholding) is not warranted. There are newer approaches (e.g. Marriott Haresign et al., 2022: <https://www.sciencedirect.com/science/article/pii/S1878929321001146#fig0015>)(<https://www.sciencedirect.com/science/article/pii/S1878929321001146#fig0015>)) for ICA in infants, which also demonstrate a relatively high error rate for MARA (38%). Going forward I would like to encourage the authors to provide an analysis either with no ICA correction or with manually selected ICA components.

With regards to the FOOOF parametrization: The authors should better explain what were the modifications of the SpecParam? and why did the modification affect mainly the main signal of interest (low beta). It is interesting to see that the biggest SpecParam error is exactly in the range of the signal of main interest (low beta), which can be problematic. Finally, why was the

subplot C not conducted for all the age bins? This should be added. As a side note, the reference to supplementary figure 4 is wrong. It should be supplementary figure 7.

Supplementary Figure 8 should be reorganized to make the behavioral scores better comparable. Thus, please provide a separate figure for each age bin (rather than behavioral score) in which the different behavioral scores are depicted in one subplot. In addition, also provide this information for broader age range (not just 12 -36 months).

With regard to the stability of (anesthesia-induced) alpha coherence during development, the authors reference to another publication (Cornelissen et al. 2018). The authors should provide this information on their own data set. Figure 1G indicates an increase in the alpha peak with development.

Furthermore the authors slightly misunderstood my comment regarding the artifact free segments. My point was that there were (most probably) subjects that had multiple 30-seconds artifact free segments. Thus, this might result in different numbers of artifact free segments per subject and thus provide different SNRs, no?

Supplementary Figure 6: I don't think it is a valid approach to create topoplots from 6 electrode clusters rather than based on each electrode. This provides very biased topoplots due to smoothing artifacts. Please provide topoplots which are based on the 128 electrodes or 63 electrodes (or downsample everything to 64 electrodes). Also what was the idea to create what is the motivation for the colorbar? For power, one should use a unimodal color scale (e.g. white to red). Also, what is the reason that gamma power is has negative values. please provide labels for the colorbar.

My initial comment: "Finally, analysis for low beta peak association with anesthesia-induced frontal alpha coherence: The analyses for this research question are vastly unclear from the results section. It is unclear why only 23 infants (indicated in the figure) were analyzed from the 36 recorded infants (30% exclusion rate seems to be very high)."

was not addressed by the authors.

With regard to my comment to account for sex and age and thus not to conduct an ANCOVA but rather a generalized linear mixed effect (I have initially suggested a logistic regression). How do the authors justify an ANCOVA, which can be sensitive (i.e. variable results) to small samples?

The authors added additional analyses for "by 6-months only 15% of infants have two peaks in this range, and for most infants it is the higher 9.5Hz peak that is no longer observed. At 6-months fewer than 40%, of infants exhibit a dominant peak in the "alpha" (6.5-12Hz) range (Fig 1D) and the average peak frequency in the theta/alpha range is 6.3 ± 1 Hz. This disappearance of the higher peak after 4-months of age may reflect a transient step in thalamocortical circuitry development."

However, the reported results are not in line with common good practices of reporting statistical results (only providing p-values)

With regard to my comment: "I would recommend not to use "aperiodic power" as terminology. The better terminology is aperiodic". The authors stated that they changed the terminology, but this is not the case for the title, abstract, and introduction.

Finally, why is the Rico-Pico et al., 2023 study not cited in the manuscript, although mentioned several times in the response and highly relevant to this work? Again, an elaboration on what the present paper provides more information than the Rico-Pico paper should be provided.

Reviewer #3 (Remarks on code availability):

I have downloaded the analysis code from OSF and have peeked into some of the files. But, I did not try to reproduce the results. However I did not find a README file with a sufficient amount of instructions to install and run the analysis. In general the analysis can not be reproduced because, the data is not made available (the authors statement: Consents obtained from human participants prohibit sharing of de-identified individual data without data use agreement in place. Please contact the corresponding author with reasonable data requests.).

Nature Communications NCOMMS-23-37310-B

Developmental trajectories of EEG aperiodic and periodic power: Implications for understanding thalamocortical development during infancy.

Dear ,

We thank the reviewers for their feedback on our resubmission. Reviewer comments are highlighted in gray with our responses below.

Reviewer #1 (Remarks to the Author):

The authors have done a very thorough job at responding to our comments and recommendations. We appreciate the softened language regarding the maturation of thalamocortical development, and find this to be a sufficient change. Also, the two comments we made regarding the statistical models — ordered effect of sex and within-subjects analyses — totally make sense. Those were silly oversights on our parts regarding the mixed effect models used.

Reviewer #1 (Remarks on code availability):

Because everything is organized into numbered computational notebooks, it was easy to follow without an explicit README. Everything looks straightforward.

We thank Reviewer 1 for their constructive feedback during the review process.

Reviewer #2 (Remarks to the Author):

I appreciate the authors for being responsive to the reviewers' feedback. I think the manuscript has been greatly improved. I only have a couple minor comments.

1. For the supplemental analyses involving motion, it would be important to add details on how many infants and EEG sessions were coded – was it all of the data for only one study? It would also be important to provide more details on the coding scheme that was followed by the research assistants as well as any details on the training and reliability of the coding scheme.

We appreciate Reviewer #2 overall positive response. The data related to behavioral codes was only available for one study (Study 4). We have added information regarding the numbers of infants at each age with each behavioral code, as well as information regarding the coding protocol in the supplemental materials. Research assistants who provided behavioral codes underwent a training procedure where they were observed by a senior research assistant in all aspects of EEG acquisition and needed to demonstrate reliability over 3 sessions before being independent. However, there was not a process in place to check for inter-rater reliability of coding. The table below has been added to the supplemental figure.

Number of participants

Behavior Score	2m	12m	18m	24m	36m
2 Frequently upset, only short periods of calm between agitation	0	44	28	22	14
3 Sometimes upset (30-50% of session) with long periods of calm between protests	13	69	42	25	29
4 Rarely upset with agitation/upset 10-30% of session	17	54	39	33	24
5 Calm for majority (90%) of session	10	22	27	18	23

2. For the anesthesia analyses, I still think it is confusing to mention on p. 22 a baseline period (pre-anesthesia) that was never analyzed in the current manuscript, but then mention that “baseline EEG” were preprocessed and analyzed for that analysis. For example, “Baseline EEGs (n = 45) were visually inspected and...” Please clarify in the manuscript that the baseline period was not analyzed and only the period during anesthesia was utilized in the analyses presented in manuscript.

We apologize for the confusion and the misunderstanding regarding our previous response - the baseline period (pre-anesthesia) **was** analyzed as part of Figure 4. To clarify, (1) baseline (pre-anesthesia) EEG data AND (2) EEG data following anesthetic administration were both analyzed, but in different ways. We hypothesized that the presence of a low beta peak in the baseline EEG reflects maturation in thalamocortical connectivity and wanted to test this against a functional measure of thalamocortical maturation – anesthesia-induced alpha coherence.

The power spectra from baseline (pre-anesthesia) data are shown in Figure 4a. Analysis of this baseline data determined whether infants had a low beta peak. To test the above hypothesis, we then assessed whether infants with presence of a low beta peak in their baseline EEG had higher anesthesia-induced alpha coherence (measured during anesthesia). We have made changes to the results section and methods to improve clarity (Page 11, Page 23 line 31).

3. The authors should also cite the recent manuscripts (Rayson et al., 2023; Rico-Pico et al., 2023) that find a similar pattern of results, as they do in the letter.

4. Similarly, in the letter, the authors mention adding a limitation on p. 16, which is not in the revised manuscript. Specifically, the following paragraph (mentioned in the letter) is not present in the manuscript: “Third, the findings presented here utilize a specific artifact removal and...” Please add that entire section to the manuscript.

Thank you for catching this oversight. We certainly meant to include the paragraph in the limitations and the associated citations.

Reviewer #3 (Remarks to the Author):

I would like to thank the authors for addressing my and the other reviewers comments. Some of the (others and mine) comments were answered correctly and provided good explanation. However, other responses do not provide sufficient justification or evidence to the initial comments. In the following address them specifically:

I thoroughly agree with the reviewer #1 in terms of “But given it is the primary argument of the paper, it would be nice to have another dataset that could be used to validate the larger hypothesis, or to soften the language on this topic.”

The authors softened the language, but the statements regarding the thalamocortical connectivity are still purely speculative. The data and results this study simply do not provide this information. Thus such claims in the title and abstract are not valid.

Abstract:

“We present preliminary evidence that the emergence of the low beta peak is associated with higher thalamocortical-dependent, anesthesia-induced alpha coherence”

Introduction:

“Consistent with our hypothesis, we find infants with an identifiable low beta peak have higher anesthesia-induced alpha coherence than those that do not, providing preliminary evidence that the emergence of this peak may be associated with thalamocortical loop maturation.”

Discussion:

“While we provide some preliminary evidence that the emergence of low beta is associated with maturation of thalamocortical connectivity, the dataset is small and still an indirect measure.”

It’s not even about the small sample size, which is a major problem as I stated in my initial reviews, but rather that there might be several competing alternative explanations of the emergence of low beta rather than maturation of thalamocortical connectivity. Such a claim is only warranted if additional data (e.g. DTI) would demonstrate such a relationship (e.g. as in Troendle et al. 2022: <https://elifesciences.org/articles/77571>).

We thank the Reviewer for this feedback and appreciate the point raised. We have made significant edits to both our title, abstract, and introduction (page 4).

Title is now: Developmental trajectories of EEG aperiodic and periodic components in children 2-to-44 months of age.

Abstract: Changes were made to better describe the hypothesis being examined with the anesthesia cohort data and state the findings without claiming they directly support thalamocortical loop maturation.

Introduction: The sentence highlighted by the reviewer above has been removed. In addition, within the introduction (page 4 – line 23-25) and results section (page 11) we have also included additional

background information to clarify our hypothesis and current evidence that anesthesia-induced alpha coherence is dependent on thalamocortical connectivity, supporting its use as an indirect, but valid, measure of thalamocortical connectivity. Computational modeling (Ching et al 2010) has suggested that anesthesia-induced coherent frontal alpha oscillations are generated in the thalamus and reliant on mature connections between the thalamus and cortex. This has been directly tested through in vivo recordings of Sprague-Dawley rats with electrodes in both medial prefrontal and thalamic nuclei during induction of unconsciousness using propofol (Flores et al. PNAS 2017). EEG dynamics observed during anesthesia induction in rats were similar to what has been observed in humans with increased beta oscillations, transitioning to alpha oscillations at the time of loss of consciousness. They then analyzed coherence between six different pairs of thalamic nuclei and cortical layers, and observed significant alpha coherence specifically during loss of consciousness (loss of righting reflex), but not prior to anesthesia. Simultaneous recordings of thalamic nuclei and cortices have also been performed in monkeys during the [propofol induced] unconscious state, observing “sustained increase in alpha (7-15Hz) frequency range phase synchrony while Unconscious” (Bastos et al. 2021).

We have added clarification regarding this hypothesis to page 11:

“Multiple lines of evidence suggest that anesthesia-induced frontal alpha coherence is dependent on thalamocortical connectivity^{38,42,43}, and in infants robust levels of alpha coherence with anesthesia administration are not observed until 10 months of age⁴⁰. We hypothesized that the emergence of low beta oscillations (as measured by the presence of a low beta peak) beginning after 7-months of age may reflect maturation of the thalamocortical loop also responsible for the developmental emergence of anesthesia-induced frontal alpha coherence around the same age⁴⁰. To explore this possibility, we assessed infants participating in cross-sectional study where EEG data was collected before and during exposure to GABA-modulating sevoflurane anesthesia⁴⁶.”

We have made sure in the limitations section to state that our findings are preliminary, that the dataset is small, and that anesthesia-induced alpha-coherence is an indirect measure of thalamocortical connectivity. We also state that *“Future research that combines EEG and MRI in infants from 4-6 months and again at 10-15 months, could provide more direct evidence linking alpha and beta periodic EEG findings with the development of thalamocortical connectivity.”*

Ching S, Cimenser A, Purdon PL, Brown EN, Kopell NJ. Thalamocortical model for a propofol-induced alpha-rhythm associated with loss of consciousness. Proc Natl Acad Sci U S A. 2010 Dec 28;107(52):22665-70. doi: 10.1073/pnas.1017069108. Epub 2010 Dec 13. PMID: 21149695; PMCID: PMC3012501.

Flores FJ, Hartnack KE, Fath AB, Kim SE, Wilson MA, Brown EN, Purdon PL. Thalamocortical synchronization during induction and emergence from propofol-induced unconsciousness. Proc Natl Acad Sci U S A. 2017 Aug 8;114(32):E6660-E6668. doi: 10.1073/pnas.1700148114. Epub 2017 Jul 25. PMID: 28743752; PMCID: PMC5558998.

André M Bastos, Jacob A Donoghue, Scott L Brincat, Meredith Mahnke, Jorge Yanar, Josefina Correa, Ayan S Waite, Mikael Lundqvist, Jefferson Roy, Emery N Brown, Earl K Miller (2021) Neural effects of propofol-induced unconsciousness and its reversal using thalamic stimulation eLife 10:e60824

Moreover, the response to reviewer #1 valid suggestion to analyze the data in a within-subject design surprised me quiet a bit: "While our dataset is longitudinal, our ability to examine developmental trajectories in an exclusively within-subjects fashion is limited by the fact that many participants do not have data from all timepoints." From Figure 1C, I speculate that the majority of the sample has at least 4 time points, which would qualify such a within-subject analysis.

Unfortunately, the majority of the sample does not have at least 4 time points. As Figure 1C shows, every participant in Study 3 (which makes up 61% of participants) had a maximum of two time points. As described in Table 1, the average number of data points across the full data set is 3.2. We also note that for those infants with at least 4 time points, an even smaller portion overlap in exact time points. In order to leverage all the data provided we use a repeated measures GAMMs analysis that includes within subject data and thus is not entirely between subjects.

Reviewer 3 raised several concerns regarding the automated preprocessing pipeline used in this paper. We have combined these comments below in order to address them in an organized way.

To the comment of reviewer #2: "I believe the authors are using previous versions of the HAPPE software to clean their data. The old version is known to overcorrect during artifact rejection, potentially taking away aspects of the signal. Why did the authors decide to use this method over the new version of the method." The authors responded that "Importantly, in our testing of HAPPE V3, we have found that model fits were poor using SpecParam, even after utilizing our edited code, preventing a direct comparison of values used in the manuscript. Given, this we proceeded with using HAPPE 1.0." How such a discrepancy between the two pipelines in terms of SpecParam fit is explainable? As the reviewer #2 correctly states is that the HAPPE software overcorrects the data (which I also stated in my initial review with regard the the % of bad ICA by MARA).

The reasoning about the preprocessing (ICA MARA, HAPPE pipeline) are further not convincing, mainly because of the validation approach in the Gabard-Durnam et al 2018 paper is rather inaccurate, as they do not have a true ground truth what the EEG signal is and thus can not really claim that % EEG signal retained from 25-70%.

Moreover, some of the authors on this paper are also authors of the HAPPE pipeline and thus an independent evaluation of the MARA (preceding wavelet thresholding) is not warranted. There are newer approaches (e.g. Marriott Haresign et al.,

2022: <https://www.sciencedirect.com/science/article/pii/S1878929321001146#fig0015> (<https://www.sciencedirect.com/science/article/pii/S1878929321001146#fig0015>)) for ICA in infants, which also demonstrate an relatively high error rate for MARA (38%). Going forward I would like to encourage the authors to provide an analysis either with no ICA correction or with manually selected ICA components.

We agree that there is vigorous debate about which artifact removal pipelines should be used, but there is no consensus or a gold standard in place. A detailed comparison of HAPPE 1.0 vs HAPPE 3.0 is beyond the scope of this paper, which is focused on developmental trajectories of EEG power and not comparing pipelines. Since its publication, HAPPE 1.0 has been widely used for many studies of infant and child brain development. This includes work from the NIH-funded Autism Biomarker Consortium for Clinical Trials (ABC-CT), consortium whose goal is to evaluate the stability and reproducibility of resting state EEG measures (Webb et al 2022). Reviewer 3 is correct that we are most experienced using the HAPPE 1.0 pipeline (the first author of this paper was involved in testing the HAPPE 1.0 pipeline on infant data), and because of this experience we are confident in the reliability and reproducibility of the EEG outputs from this particular processing pipeline. Our decision to use the BEAPP/HAPPE1.0 pipeline was also influenced by the Autism Biomarker Consortium's work in assessing EEG reliability and reproducibility for future FDA approval. We felt it was beneficial to use the same pipeline as ABC-CT, as this would allow for more direct comparison with future FDA approved biomarkers. However, we also do not wish to ignore the reviewer's concerns regarding over-correction during artifact rejection due to ICA correction with MARA. We have performed several additional analyses with a set of 9 EEG files, comparing the HAPPE1.0 pipeline with and without ICA with MARA. Both analyses include the original HAPPE1.0 wavelet-enhanced thresholding step. Results are shown on the next page for Absolute (top) and periodic (bottom) power for each individual EEG. Absolute power spectra and periodic power spectra were very similar at the individual level, suggesting that the ICA-MARA step did *not* have a significant influence on EEG power spectral features. We value the additional ICA-MARA step in our pipeline as it provides additional quality metrics of the data retained and have kept in our analyses for this paper.

We realize that in our previous response to Reviewer 2, we had written a lengthy limitation regarding processing pipelines but had accidentally not transferred it to the final draft of the paper. This has been added (Page 16-17):

“Fourth, the findings presented here utilize a specific artifact removal and processing pipeline (BEAPP and HAPPE1.0). An automated pipeline was used to improve reproducibility of our findings, and the code is available on Open Science Framework (<https://osf.io/u3gp4>). We do note that similar developmental changes in the low and high beta from 9 and 12 months are shown in Figure 1 of Rayson et al. 2023⁷³ which used a combination of MADE and NEAR pipelines. Interestingly, recently Rico-Pico et al. 2023⁷⁴ also described developmental changes in the periodic power spectra using MADE processing pipelines but limited their analysis to 1-20Hz “due to a power bump” in the gamma range, assumed to be related to muscle artifact, but likely the same high beta peak we observe most prominent at 7-8 months of age. Power spectra from 6, 9, and 16 months shown in Supplemental Figure 8 of Rico-Pico et al. 2023 visually show a similar pattern in beta band change across development. We hope that our comprehensive characterization of developmental changes using a single pipeline with frequent age sampling across the first 3 years of development spur other infant EEG researchers to evaluate the patterns we describe in their own data.”

Finally, we are reassured by the signal quality and reliability provided by our EEG collection and processing pipeline for several reasons. (1) The developmental changes observed are visualized both on group average and individual EEGs (eg Supplemental Figure 2 – Age 2 months vs 4 months – loss of alpha peak; 6 months vs 18 month, loss of beta “trough”). (2) As we discussed in our previous response, using this data we can accurately predict an infant’s chronological age within 92 days. This suggests that EEG signal collected and computed on the individual level consists of biological meaningful data as it relates to brain development. (3) At the individual level we observe that while periodic peak characteristics change with the age, they also maintain their individual peak characteristics. Several examples of 9, 12, and 18 month periodic power spectra from the same infants are shown below. Visually, the shape and peak frequency of both alpha and high beta peaks are remarkably similar across individual power spectra measured 3-6 months apart, but are different across individuals. Poor signal quality or excessive noise would prevent these similarities on the individual level from being observed. While it is possible that the BEAPP/HAPPE 1.0 pipeline “overcorrects” and removes more signal than other pipelines, we believe our data support that the signal retained is biologically meaningful and reproducible, and the findings presented are the result of developmental changes in brain activity and not artifact.

References

Webb SJ, Naples AJ, Levin AR, Hellemann G, Borland H, Benton J, Carlos C, McAllister T, Santhosh M, Seow H, Atyabi A, Bernier R, Chawarska K, Dawson G, Dziura J, Faja S, Jeste S, Murias M, Nelson CA, Sabatos-DeVito M, Senturk D, Shic F, Sugar CA, McPartland JC. The Autism Biomarkers Consortium for Clinical Trials: Initial Evaluation of a Battery of Candidate EEG Biomarkers. *Am J Psychiatry*. 2023 Jan 1;180(1):41-49. Doi: 10.1176/appi.ajp.21050485. Epub 2022 Aug 24.

Brandes-Aitken A, Metser M, Braren SH, Vogel SC, Brito NH. Neurophysiology of sustained attention in early infancy: Investigating longitudinal relations with recognition memory outcomes. *Infant Behav Dev*. 2023 Feb;70:101807. Doi: 10.1016/j.infbeh.2022.101807. Epub 2023 Jan 10. PMID: 36634407; PMCID: PMC9901300.

I have used the Geodesic NetAmps for >10 years and I know using a impedance threshold below 100kOhm would not provide us good quality of data even in healthy children and adults, for which movement artifacts are less of a problem as in infants. While I understand that adjusting impedences makes the infants become fuzzy, the signal of interest is very sensitive to good signal quality, which in my opinion is not given in this data set.

Infant researchers using EGI/MagStim amplifiers frequently use an impedance threshold of 100k Ω . Please see example references below (Debnath et al 2020; Marin et al 2020). We have added the following as a limitation: *“Third, impedance thresholds were kept below 100k Ω , rather than 50 k Ω , in order to reduce the time needed to optimize impedance and in turn prevent infants and toddlers from becoming fussy. While all data were collected in electrically-shielded rooms, it is possible that recordings with impedances above 50 k Ω have reduced signal quality. However the developmental changes described are robust and observed across individuals as observed in Supplemental Figure 2.”*

Debnath R, Buzzell GA, Morales S, Bowers ME, Leach SC, Fox NA. The Maryland analysis of developmental EEG (MADE) pipeline. *Psychophysiology*. 2020 Jun;57(6):e13580. doi: 10.1111/psyp.13580. Epub 2020 Apr 15. PMID: 32293719.

Marin A, Hutman T, Ponting C, McDonald NM, Carver L, Baker E, Daniel M, Dickinson A, Dapretto M, Johnson SP, Jeste SS. Electrophysiological signatures of visual statistical learning in 3-month-old infants at familial and low risk for autism spectrum disorder. *Dev Psychobiol*. 2020 Sep;62(6):858-870. doi: 10.1002/dev.21971. Epub 2020 Mar 25.

With regards to the FOOOF parametrization: The authors should better explain what were the modification of the SpecParam? and why did the modification affected mainly the main signal of interest (low beta). It is interesting to see that the biggest SpecParam error is exactly in the range of the signal of main interest (low beta), which can be problematic. Finally, why was the subplot C not conducted for all the age bins? This should be added. As a side note, the reference to supplementary figure 4 is wrong. It should be supplementary figure 7.

We apologize for confusion and hope to clarify here.

As described in the methods, page 21-22, the modifications of SpecParam were implemented because there was poor model fit in the low beta range (10-20Hz) using the original SpecParam. The increased error in this range occurs because there is a “trough” in the power spectra (green) between 10-20Hz at younger ages (Supplemental Figure 7A; 5 month spectra shown to the right). For many infants in this age range, the first estimated ap_fit (blue) defined by the `robust_ap_fit` function, “cut through” the original power spectra and based on the original SpecParam code, this then led the modeled spectrum (red) to be higher in power than the original spectrum (see Figure to the right). We describe reasons for this below, as well as the changes made to the code. Changes to code are also documented in the code available on osf.io/u3gp4 and written so that they are diffable with the original SpecParam code, with links provided.

In the original `robust_ap_fit` function (<https://github.com/foof-tools/foof/blob/5e655d73c9d7a47d0411b5177657aaed67c69d6d/specparam/objs/fit.py#L964-L1024>), the first estimated flatspec was calculated by subtracting the initial ap fit from the power spectra, AND any value below 0 was converted to 0 (line 993). This leads to the increased error between the original spectrum and the foof estimated spectrum in the 10-20Hz region for 2-7 month olds, since many values fell below 0. Our modified code ‘new_robust_ap_fit’ elevates the first estimated flatspec such that the lowest point in the flatspec is ≥ 0 .

```
# OLD: Flatten outliers, defined as any points that drop below 0
# flatspec[flatspec < 0] = 0 #ORIGINAL
# NEW: Increase baseline to prevent fitting negative values
if min(flatspec) < 0:
    flatspec -= min(flatspec)
```

Following this step, a second more robust aperiodic fit is estimated, and then the fit function re-estimates the flattened spectra (spectrum_flat). The original fit function found here:

<https://github.com/foof-tools/foof/blob/5e655d73c9d7a47d0411b5177657aaed67c69d6d/specparam/objs/fit.py#L427-L525>.

Our modified code then sets any negative data in the flattened spectra equal to 0 (similar to the approach used in the original code during the initial aperiodic fit).

```
# Fit the aperiodic component
self.aperiodic_params_ = self._robust_ap_fit(self.freqs, self.power_spectrum) #ORIGINAL
self.aperiodic_params_ = self._new_robust_ap_fit(self.freqs, self.power_spectrum) #NEW
self._ap_fit = foof.sim.gen.gen_aperiodic(self.freqs, self.aperiodic_params_) #edited to make standala

# Flatten the power spectrum using fit aperiodic fit
self._spectrum_flat = self.power_spectrum - self._ap_fit

self._spectrum_flat[self._spectrum_flat < 0] = 0 #NEW
```

Regarding Subplot C– the error in the FOOOF vs original Spectra fit was highest for 3-7 months for the original SpecParam code, and so we wanted to show this in more detail by graphing the squared error across the frequencies. We have now created a separate Supplemental Figure 8, that shows the squared error for all the age bins. As described above, the squared error is substantially reduced after 7 months, especially when using our modified code (shown in blue, Supplemental Figure 8). We would like emphasize that the increased error in the low beta range occurs because there is a “trough” (ie lack of a beta peak) that is not accurately captured in the original SpecParam code, leading to significant differences between the fofof estimated spectra (in green) and the actual spectra (red). The modifications to the SpecParam code were necessary in order to accurately capture developmental changes in this frequency range, and the low error in the modified code, especially after 7 months, supports the accuracy of our low beta periodic peak measurements.

Supplemental Figure 8: Comparison of squared error across frequencies based of unedited (orange) and modified (blue) SpecParam estimates of whole brain power.

Supplementary Figure 8 should be reorganized to make the behavioral scores better comparable. Thus, please provide a separate figure for each age bin (rather than behavioral score) in which the different behavioral scores are depicted in one subplot. In addition, also provide this information for broader age range (not just 12-36 months).

Thank you for this suggestion. We have added a separate figure based on age bin and also provided the sample size for each category. As we describe in the supplemental figure legend, behavioral scores were only completed for Study 4 and were not available for other studies. Study 4 only includes the following ages: 3m, 12m, 18m, 24m, and 36m. We have now added the 2-3m data as well, which we had previously not included as it was more limited in sample size and did not have any behavioral codes below 3.

With regard to the stability of (anesthesia-induced) alpha coherence during development, the authors reference to another publication (Cornelissen et al. 2018). The authors should provide this information on their own data set. Figure 1G indicates an increase in the alpha peak with development.

The data shown in Figure 1G reflects power (not coherence) captured under the awake settings from our large longitudinal sample and not in the cohort of children used for our anesthesia analysis, and thus is not a direct comparison of developmental changes in anesthesia-induced alpha coherence. The sample used to assess anesthesia-induced alpha coherence is a separate sample described on page 23 (Anesthesia Cohort), much smaller in size prohibiting an accurate analysis of developmental

changes in alpha coherence. Instead we reference the Cornelissen et al 2018 paper regarding stability of anesthesia alpha coherence as it consists of a larger sample across a broader age range (n=91). In Figure 4B (shown below), they show developmental changes in alpha **coherence** during anesthesia. There is a clear lack of alpha coherence early in development (<7 months) and once infants have alpha coherence (~9-10 months), the peak frequency is stable between 8-12Hz. Our data is purposefully limited to a developmental period where infants may or may not exhibit significant alpha coherence due to individual variability in brain development. Identifying a “peak” for individuals without significant alpha coherence would not be meaningful. Given this, we are using the literature and a study sample much larger than our own to establish or analysis protocol.

Furthermore the authors slightly misunderstood my comment regarding the artifact free segments. My point was that there were (most probably) subjects that had multiple 30-seconds artifact free segments. Thus, this might result in different numbers of artifact free segments per subject and thus provide different SNRs, no? With regard to my comment to account for sex and age and thus not to conduct a ANCOVA but rather a generalized linear mixed effect (I have initially suggested a logistic regression). How do the authors justify an ANCOVA, which can be sensitive (i.e. variable results) to small samples?

Thank you for this suggestion of using a linear mixed effect model. We agree that variability in the number of artifact free segments across participants could lead to different SNR across participants. Using an LME allowed us to include all available artifact-free segments, without casewise deletion, as LMEs account for different numbers of segments through partial pooling of the model’s variance (Heise *et al.* 2022). In this method, infants with fewer artifact-free segments are weighted less in the group mean, than those with more segments – better accounting for differences in SNR, while also retaining as many infants as possible.

Using this analysis, we find that there is a significant association between the presence of a low beta peak and median alpha coherence (LME $\beta = 0.12$, SE = 0.051, $p = 0.02$). As with the anova analysis, sevoflurane level was included as a covariate ($\beta = - 0.055$, SE = 0.020, $p = 0.006$). We have added these results to the manuscript. We also note that there was no significant differences in the number of artifact free segments between infants with (mean = 6.4, SD = 4.0) and without (mean = 7.5, SD= 4.7) a low beta peak.

Heise MJ, Mon SK, Bowman LC. Utility of linear mixed effects models for event-related potential research with infants and children. *Dev Cogn Neurosci.* 2022 Apr;54:101070. doi: 10.1016/j.dcn.2022.101070. Epub 2022 Jan 15. PMID: 35395594; PMCID: PMC8987653.

Supplementary Figure 6: I don't think it is a valid approach to create topoplots from 6 electrode clusters rather than based on each electrode. This provides very biased topoplots due to smoothing artifacts. Please provide topoplots which are based on the 128 electrodes or 63 electrodes (or downsample everything to 64 electrodes). Also what was the idea to create what is the motivation for the colorbar? For power, one should use a unimodal color scale (e.g. white to red). Also, what is the reason that gamma power is has negative values. please provide labels for the colorbar.

Thank you for this feedback. We were also were hesitant to make topoplots in this way due to the smoothing artifacts, but wanted to respond to reviewer recommendations in our first revision. In reviewing original comments from reviewers, our understanding is that the goal topographic visualization is to better visualize the region of interest GAMMS model estimates shown in Supplemental Figure 5, as these account for both the within subject longitudinal data, as well as sex and study factors. As our GAMMs model estimates are based on measures calculated from SpecParam modeled outputs based on power spectral averages across electrode clusters we felt that showing topoplots based on 128 or 64 electrodes clusters would not accurately represent the model-based estimates. To provide a more accurate representation of the modeled data but displayed topographically, we have created a new Supplemental Figure 6 that shows the 12 EEG measures modeled in the paper by the 4 regions of interests in a unimodal color scale (white to red). Portions of the figure are shown below:

Supplemental Figure 6

My initial comment: "Finally, analysis for low beta peak association with anesthesia-induced frontal alpha coherence: The analyses for this research question are vastly unclear from the results section. It is unclear why only 23 infants (indicated in the figure) were analyzed from the 36 recorded infants (30% exclusion rate seems to be very high)." was not addressed by the authors.

We addressed this in our original response with clarification in the text. We have made additional clarifications in the text. The reduction from 36 recorded infants to 23 recorded infants was due to restricting our analysis to infants who were 7-12 month old – the age range where there is variability in whether an infant will or will not have a low beta (beta 1) peak. Including the full age range would lead to problems with multicollinearity as age was associated with both presence of a low beta peak and alpha coherence (See Figure to the right). By restricting our analysis to an age where there was no association between age and low beta peak, we could be more confident that differences in alpha coherences between infants with and without a low beta peak were not due to age associations.

The authors added additional analyses for “by 6-months only 15% of infants have two peaks in this range, and for most infants it is the higher 9.5Hz peak that is no longer observed. At 6-months fewer than 40%, of infants exhibit a dominant peak in the “alpha” (6.5-12Hz) range (Fig 1D) and the average peak frequency in the theta/alpha range is 6.3±1Hz. This disappearance of the higher peak after 4-months of age may reflect a transient step in thalamocortical circuitry development.”

However, the reported results are not in line with common good practices of reporting statistical results (only providing p-values)

We have added an odds ratio to this result (page 7).

With regard to my comment: “I would recommend not to use “aperiodic power” as terminology. The better terminology is aperiodic”. The authors stated that they changed the terminology, but this is not the case for the title, abstract, and introduction.

We have updated the title, abstract, and introduction.

Finally, why is the Rico-Pico et al., 2023 study no cited in the manuscript, although mentioned several times in the response and highly relevant to this work? Again, a elaboration on what the present paper provides more information than the Rico-Pico paper should be provided.

We apologize for this oversight. This was described in a newly written limitations section to be added to the discussion, but we inadvertently did not copy into the final draft. We definitely meant for the following to be included in the original resubmitted draft. The addition to the limitations section is provided below.

“We do note that similar developmental changes in the low and high beta from 9 and 12 months are shown in Figure 1 of Rayson et al. 2023⁷³ which used a combination of MADE and NEAR pipelines.

Interestingly, recently Rico-Pico et al. 2023⁷⁴ also described developmental changes in the periodic power spectra using MADE processing pipelines but limited their analysis to 1-20Hz “due to a power bump” in the gamma range, assumed to be related to muscle artifact, but likely the same high beta peak we observe most prominent at 7-8 months of age. Power spectra from 6, 9, and 16 months shown in Supplemental Figure 8 of Rico-Pico et al. 2023 visually show a similar pattern in beta band change across development. We hope that our comprehensive characterization of developmental changes using a single pipeline with frequent age sampling across the first 3 years of development spur other infant EEG researchers to evaluate the patterns we describe in their own data.”

Reviewer #3 (Remarks on code availability):

I have downloaded the analysis code from OSF and have peeked into some of the files. But, I did not try to reproduce the results. However I did not find a README file with a sufficient amount of instructions to install and run the analysis. In general the analysis can not be reproduced because, the data is not made available (the authors statement: Consents obtained from human participants prohibit sharing of de-identified individual data without data use agreement in place. Please contact the corresponding author with reasonable data requests.).

We have added a README file to the OSF code.

Reviewer #3 (Remarks to the Author):

I appreciate the authors for being responsive to the reviewers' feedback. I have some more comments, which I would like to be addressed by the authors:

Abstract:

"We hypothesized that the emergence of the low beta peak may reflect maturation of thalamocortical network development."

Why is this sentence in past tense? It seem as the authors hypothesize this and then investigate this hypothesis. In fact this is a speculation of the authors (and should be rather stated in this way and removed from the abstract).

The same applies to the statement (page 11): " We hypothesized that the emergence of low beta oscillations (as measured by the presence of a low beta peak) beginning after 7-months of age may reflect maturation of the thalamocortical loop also responsible for the developmental emergence of analyzed data from infants participating in cross-sectional study where EEG data was collected beforeand during exposure to GABA-modulating sevoflurane anesthesia."

The analyses of the present paper do not allow to make conclusions on thalamocortical loops. This is speculations and should be stated this way.

The authors wrote: "Consistent with known increases in brain volume and synaptogenesis, we observe rapid increases in aperiodic offset in the first year, with minimal change between 1 to 3 years." This requires a reference.

The authors responded to my suggestion to conduct a within-subject analyses by:

"Unfortunately, the majority of the sample does not have at least 4 time points. As Figure 1C shows, every participant in Study 3 (which makes up 61% of participants) had a maximum of two time points."

I would still like to see the within subject analyses for those infants that have the >3 timepoints available.

The authors did not really addressed my previous comment: "The reasoning about the preprocessing (ICA MARA, HAPPE pipeline) are further not convincing, mainly because of the validation approach in the Gabard-Durnam et al 2018 paper is rather inaccurate, as they do not have a true ground truth what the EEG signal is and thus can not really claim that % EEG signal retained from 25-70%." This is a substantial problem of the HAPPE pipeline. Moreover, the MARA approach is concerning and should be replicated with a different approach (as I have suggested in my previous round of review). The authors provided visuals for 9 EEG files, which were re-assessed for with and without ICA MARA. This is not really informative, because 1) arbitrary 9 files 2) based on this, we can not assess the effect on the measures of interests. Thus, I would like to see the replication of the main results for the preprocessed EEG data without ICA.

" (2) As we discussed in our previous response, using this data we can accurately predict an infant's chronological age within 92 days. This suggests that EEG signal collected and computed on the individual level consists of biological meaningful data as it relates to brain development. "

This is a faulty answer. If artifacts (e.g. head movement) are related to age, then the prediction of infants chronological age would be possible purely on the head movement. Thus, it does not prove that the EEG signal consists of biologically meaningful data.

Argument 3 of the authors (intra vs. interindividual variability) is more convincing.

There are inconsistencies throughout the manuscript w.r.t. the units of aperiodic slope and intercept. Sometimes the authors use "a.u." for the slopes, sometimes micro Volt (which is wrong). The correct units for aperiodic slope should be provided ($\log(\mu\text{V}^2)/\text{Hz}$). Please also provide consistent units across the manuscript.

The authors provided a rather unusual "topoplot" for the 6 electrode cluster (Supplementary Figure 6). The plot is not really informative for the developmental changes (it seems that slope and offset are increasing in all electrode clusters similarly), I would recommend to run the GAMMS for each electrode individually and provide valid topoplots. I am aware that the main results are based on the 6 electrode clusters, but for visualization purposes, the true topoplot could be added to the supplementary.

"generalized linear mixed effects model (Odds Ratio = 0.98, $p < 0.0001$)." It's unclear why the authors just provided odds ratio and not the beta estimates from the glmms.

there seems to be a typo:
similar (Fig G; $F = 5.5$, $q < 0.01$).

Nature Communications NCOMMS-23-37310-B

Developmental trajectories of EEG aperiodic and periodic components in children 2-to-44 months of age.

Dear ,

Thank you for considering our resubmission. We have responded to Reviewer #3's comments from our last resubmission below. Comments from the reviewer are in gray and our responses are italicized.

Reviewer #3 (Remarks to the Author):

I appreciate the authors for being responsive to the reviewers' feedback. I have some more comments, which I would like to be addressed by the authors:

Abstract:

"We hypothesized that the emergence of the low beta peak may reflect maturation of thalamocortical network development."

Why is this sentence in past tense? It seem as the authors hypothesize this and then investigate this hypothesis. In fact this is a speculation of the authors (and should be rather stated in this way and removed from the abstract).

The same applies to the statement (page 11): " We hypothesized that the emergence of low beta oscillations (as measured by the presence of a low beta peak) beginning after 7-months of age may reflect maturation of the thalamocortical loop also responsible for the developmental emergence of analyzed data from infants participating in cross-sectional study where EEG data was collected beforeand during exposure to GABA-modulating sevoflurane anesthesia."

The analyses of the present paper do not allow to make conclusions on thalamocortical loops. This is speculations and should be stated this way.

The sentence in the abstract is written in the past tense to reflect the fact that this was an a priori hypothesis, which was hypothesized before the data was examined. Prior to the processing of any of the anesthesia data, Dr. Wilkinson wrote the following in email correspondence to Dr. Purdon (December 13, 2021):

"Hypothesis: If the underlying baseline power spectra curve represents developmental change in inhibitory circuitry or emergence/matruation of thalamocortical circuits, we hypothesize that the features in the baseline power spectra (such as the emergence of a beta 1 peak around 12-18 month or

reduction in the beta 2 peak at 24-26 months) will be predictive of an individual's response to anesthesia (better for example than predicting based on age).

To do this, we propose evaluating pre-anesthesia vs during-anesthesia data from a range of infants who show baseline data and different developmental stages. My baseline data (see below) suggests that children between 250-450 days old (9 and 12 month age-bins) may be ideal as there is a decent amount of individual variability during this period."

Therefore, it is accurate to state that we hypothesized this relationship between low beta peak emergence and thalamocortical network development before any results were seen. We also argue that the sentence is important to keep in the abstract as it would be very unclear why we did an analysis of EEG collected under anesthesia without stating the a priori hypothesis.

Regarding any conclusions made in our paper regarding thalamocortical connections and emergence of a low beta peak, we have edited the following sentence:

"While we provide some preliminary evidence that the emergence of a low beta peak is associated with maturation of thalamocortical connectivity, the dataset is small and still an indirect measure, and thus no conclusion can be made at this time."

The authors wrote: "Consistent with known increases in brain volume and synaptogenesis, we observe rapid increases in aperiodic offset in the first year, with minimal change between 1 to 3 years." This requires a reference.

We have added references.

The authors responded to my suggestion to conduct a within-subject analyses by:

"Unfortunately, the majority of the sample does not have at least 4 time points. As Figure 1C shows, every participant in Study 3 (which makes up 61% of participants) had a maximum of two time points."

I would still like to see the within subject analyses for those infants that have the >3 timepoints available.

There are several reasons we do not think there are additional within-subjects analyses that are appropriate for this data set beyond the GAMMs we have already performed. The inclusion of the $s(\text{age_days}, ID, bs = 'fs')$ term in the GAMM models provides a smooth term for age in days, where the smooth function can vary for each subject. This allows the model to capture potentially different patterns of the relationship between age and the outcome variable for each subject. Thus, these models are accommodating the within-subject

variability in the data. Alternate approaches would not be as powerful in leveraging the full variability within the sample for a couple of reasons. First, the features analyzed have non-linear trajectories, so a simple change-score between two timepoints does not accurately represent the change across age for an individual. Second, even for subjects with greater than 3 time points, the time points available are not the same. If we restrict the data set to those with the same time points our sample size is dramatically reduced and underpowered to make confident conclusions. To address the request for more information about within-subject change, we have included spaghetti plots for both the full data set and those participants with >3 timepoints. These plots show similar within-subject trends to the GAMM models which statistically capture within-subjects variability.

The authors did not really address my previous comment: “The reasoning about the preprocessing (ICA MARA, HAPPE pipeline) are further not convincing, mainly because of the validation approach in the Gabard-Durnam et al 2018 paper is rather inaccurate, as they do not have a true ground truth what the EEG signal is and thus can not really claim that % EEG signal retained from 25-70%.” This is a substantial problem of the HAPPE pipeline. Moreover, the MARA approach is concerning and should be replicated with a different approach (as I have suggested in my previous round of review). The authors provided visuals for 9 EEG files, which were re-assessed for with and without ICA MARA. This is not really informative, because 1) arbitrary 9 files 2) based on this, we can not assess the effect on the measures of interests. Thus, I would like to see the replication of the main results for the preprocessed EEG data without ICA.

“(2) As we discussed in our previous response, using this data we can accurately predict an infant’s chronological age within 92 days. This suggests that EEG signal collected and computed on the individual level consists of biological meaningful data as it relates to brain development.”

This is a faulty answer. If artifacts (e.g. head movement) are related to age, then the prediction of infants chronological age would be possible purely on the head movement. Thus, it does not prove that the EEG signal consists of biological meaningful data.

Argument 3 of the authors (intra vs. interindividual variability) is more convincing.

We have processed data from 54 participants with both 12 month and 24 month EEG data collected from Study 4 both with and without ICA-MARA. At the pipeline level, shapes of the periodic power spectra are very similar at both ages. Changes between 12 and 24 months [strengthening of the low beta peak, reduction in high beta peak, and shift in peak alpha frequency] are also observed in both pipelines.

Below we also show the data at the participant level a both ages (12-month data in orange and 24-month data in blue), and processed with (solid line) and without ICA-MARA (dotted line), providing additional evidence of the inter-individual vs intra-individual consistency present in our processed data between time points. To assess similarity across the spectra between processing pipelines we calculated Pearson's correlations between an individual's periodic power spectrum determine with vs without ICA-MARA. Mean Pearson R was 0.95 with SD 0.08 for 12 month EEGs, and 0.95 with SD of 0.06 for 24 month EEGs, confirming the high level of similarity between outputs. Given the consistency between data processed with and without ICA-MARA, it would be very unlikely that main findings would change.

There are inconsistencies throughout the manuscript w.r.t. the units of aperiodic slope and intercept. Sometimes the authors use “a.u.” for the slopes, sometimes micro Volt (which is wrong). The correct units for aperiodic slope should be provided ($\log(\text{microVolt}^2/\text{Hz})$). Please also provide consistent units across the manuscript.

We have kept the consistent use of a.u. for slope. Aperiodic slope is calculated in the log-log space, such that the unit would be: $[\log(\text{microvolt}^2)]/[\log(\text{Hz})]$ – different than the reviewer states. A.U. is more commonly used. We have corrected our supplemental “topoplot” visualization for aperiodic slope.

The authors provided a rather unusual “topoplot” for the 6 electrode cluster (Supplementary Figure 6). The plot is not really informative for the developmental changes (it seems that slope and offset are increasing in all electrode clusters similarly), I would recommend to run the GAMMS for each electrode individually and provide valid topoplots. I am aware that the main results is based on the 6 electrode clusters, but for visualization purposes, the true topoplot could be added to the supplementary.

The recommendation from the reviewer would require first running SpecParam models for each electrode (36 electrodes instead of 4 clusters), and then extracting each of the 12 features for each electrode, and averaging across participants. This would require substantial code revision. While this is certainly possible to do, we do not think it is scientifically warranted, as the topoplots would not accurately represent the findings presented in the main manuscript, and would not provide substantial added scientific value given that the focus of the manuscript is on changes in whole brain trajectory, and visualization of differences based on ROI are already shown in two different ways (Supplemental Figures 5 and 6).

“generalized linear mixed effects model (Odds Ratio = 0.98, $p < 0.0001$).” It’s unclear why the authors just provided odds ratio and not the beta estimates from the glmm.

This has been corrected: (Odds Ratio = 0.98, $B = -0.018$, $p < 0.0001$).

there seems to be a typo:
similar (Fig G; $F = 5.5$, $q < 0.01$).

This has been corrected: “(Fig 2G; $F = 5.5$, $q < 0.01$)”